# Revealing the hidden structure of disordered materials by parameterizing their local structural manifold

Thomas J. Hardin [1] ✉, Michael Chandross[1], Rahul Meena[2], Spencer Fajardo[3], Dimitris Giovanis[2,4], Ioannis Kevrekidis [5,6], Michael L. Falk [3,4,7,8] & Michael D. Shields [2,3,4]

Durable interest in developing a framework for the detailed structure of glassy materials has produced numerous structural descriptors that trade off between general applicability and interpretability. However, none approach the combination of simplicity and wide-ranging predictive power of the lattice-grain-defect framework for crystalline materials. Working from the hypothesis that the local atomic environments of a glassy material are constrained by enthalpy minimization to a low-dimensional manifold in atomic coordinate space, we develop a generalized distance function, the Gaussian Integral Inner Product (GIIP) distance, in connection with agglomerative clustering and diffusion maps, to parameterize that manifold. Applying this approach to a two-dimensional model crystal and a three-dimensional binary model metallic glass results in parameters interpretable as coordination number, composition, volumetric strain, and local symmetry. In particular, we show that a more slowly quenched glass has a higher degree of local tetrahedral symmetry at the expense of cyclic symmetry. While these descriptors require post-hoc interpretation, they minimize bias rooted in crystalline materials science and illuminate a range of structural trends that might otherwise be missed.

The ability to interrogate, understand and modify the microscopic structure of materials distinguishes modern materials scientists from artisans of old. Starting in the 1800s[1] and developing to this day, the capacity to pinpoint, classify, and count dislocations, vacancies, interstitials, grain boundaries, grain sizes, and other deviations from tabulated lattices revolutionized crystalline materials science[2], constituting one corner of the materials science tetrahedron[3]. The success of this structure-centric approach has sparked lasting interest in developing a similarly intuitive and useful framework for the structure of non-crystalline materials, which lack crystal symmetry but which

nonetheless feature structural patterns with first-order effects on properties[4,5].

While the lattice-grain-defect framework produces interpretable and widely applicable descriptions of crystalline materials, efforts to describe glassy structure have required tradeoffs between interpretability and generality. At the most-general extreme are atomic coordinates, which are high-dimensional and (in a glass) can be bewilderingly complex. At the other (interpretation-friendly) extreme is the simple question "is there a crystal lattice," which is of interest but which is inadequate to predict the variety of behavior observed within

[1]Material, Physical, and Chemical Sciences Center, Sandia National Laboratories, Albuquerque, NM, USA. [2]Department of Civil and Systems Engineering, Johns Hopkins University, Baltimore, MD, USA. [3]Department of Materials Science and Engineering, Johns Hopkins University, Baltimore, MD, USA. [4]Hopkins Extreme Materials Institute, Johns Hopkins University, Baltimore, MD, USA. [5]Department of Applied Mathematics and Statistics, Johns Hopkins University, Baltimore, MD, USA. [6]Department of Chemical and Biomolecular Engineering, Johns Hopkins University, Baltimore, MD, USA. [7]Department of Mechanical Engineering, Johns Hopkins University, Baltimore, MD, USA. [8]Department of Physics and Astronomy, Johns Hopkins University, Baltimore, MD, USA. ✉e-mail: tjhardi@sandia.gov

families of chemically similar glassy materials[4–6]. Since long-range order is absent in glassy materials, descriptors between these extremes have focused on short- and medium-range structure.

Physically-motivated scalar descriptors such as free volume[7,8] and flexibility volume[9,10] emphasize interpretability and performance on a focused set of problems (such as predicting ease of plastic deformation)[11]. Machine-learned scalar descriptors[12] have made some progress towards generalizing to multiple problems. In covalent glasses, topological constraint theory correlates coordination number with a range of mechanical and chemical properties[13,14]. However, scalar descriptors are necessarily lossy, discarding information that might be illustrative or useful in some other context.

Other descriptors aspire to much broader usefulness, attempting to distill the important structural features while discarding redundant or noisy degrees of freedom without passing judgment on any given feature's usefulness. Ideally, this would result in a description that reflects much of the complexity of a glass, but is lower-dimensional and more interpretable than raw atomic coordinates. Intermediately-lossy descriptors in this category include the statistics of rings in a covalent glass[15] and the radial distribution function[16]. More granular descriptors include the Z-clusters formalism[17] and related efficient packing theory[18,19] for metallic glasses, which categorize the first-nearest-neighbor Voronoi polyhedra in a sample and rationalize the geometric frustration in packing those polyhedra together into a solid, respectively.

Here we present a data-driven approach to describing the short-range structure of glassy materials that is more general and less lossy than the structural descriptors mentioned above, but more interpretable than raw atomic coordinates. Our approach is based on the notion that a local atomic environment (LAE) consisting of $n_{at}$ atoms can be thought of as a point in a $3n_{at} + n_{at}$-dimensional space (encoding three spatial dimensions and one chemical dimension for each atom). The ensemble of LAEs in a glassy sample, then, comprise a point cloud in that high-dimensional space. We hypothesize that enthalpy minimization loosely constrains those points onto an energetically-favorable lower-dimensional manifold in $3n_{at} + n_{at}$-space, while kinetics and entropy spread them out on that manifold. In this framing, the problem of finding a complete and parsimonious set of glassy structural descriptors is equivalent to parameterizing this manifold[20,21], through the use of manifold learning and nonlinear dimensionality reduction.

Our strategy is to:
1. sample LAEs from a material (each a point on the material's local structural manifold in $3n_{at} + n_{at}$-space),
2. quantify the difference between each pair of sampled LAEs (forming a square matrix of generalized distances between those points),
3. use that matrix of distances as input for agglomerative clustering[22] and diffusion maps[23] (well-established dimensionality reduction techniques), and
4. interpret the unlabeled classes and diffusion coordinates in terms of physical quantities.

Our approach complements recent efforts to use machine learning to understand the structure of metallic glasses and supercooled liquids in general. In ref. 24 the dimensionality-reduction method t-SNE[25] was applied to a radial-only basis function expansion of LAEs, neglecting angular information but nonetheless extracting meaningful trends. In refs. 26,27 autoencoders were used to elegantly reduce the dimensionality of rotationally invariant descriptors of angular information (Bond Order Parameters), neglecting radial information. In ref. 28 bond angle distribution and radial distribution functions were fed into a clustering algorithm that identified structural communities within an ensemble of LAEs. Finally, Coslovich et al.[29] used both angular-only Bond Order Parameters and radial+angular Smooth

Overlap of Atomic Positions (SOAP) descriptors in Principal Component Analysis and to train an autoencoder. Taken together, these studies all demonstrate the potential for unsupervised methods to extract meaningful structural information in a disordered material, and highlight the challenge of interpreting dimensionality-reduced structural representations. These studies also share the trait that information was discarded in an initial encoding step preceding dimensionality reduction: radial distributions drop angular information, while Bond Order Parameters drop radial information, are of limited use for analyzing second and higher-order shells of neighbors, and can be discontinuous with respect to atomic perturbation across a cutoff radius. Even more sophisticated descriptors like SOAP have been shown to systematically drop information[30]. Information lost in initial encoding necessarily cannot inform a low-dimensional representation. To address this shortcoming, in this manuscript we use a methodology that does not rely on an initial lossy encoding step: we measure generalized distances directly between LAEs, and then apply a dimensionality reduction method that operates on distances rather than on vector encodings. Complementing this advance, we introduce an inter-LAE generalized distance function (the Gaussian Integral Inner Product (GIIP) distance, described in the Methods section) that is complete (meaning that the generalized distance between LAEs vanishes if and only if the LAEs are identical; a dimensionality reduction based on an incomplete distance function is necessarily lossy), rotationally invariant, and continuous and smooth with respect to atomic perturbation across a cutoff radius. This combination of qualities is, to our knowledge, novel in the literature.

## Results

We applied our strategy to two samples. Our first sample is an easily understood two-dimensional model crystal with defects, which we use to illustrate both our manifold learning strategy and an approach to interpreting the data-mined parameters. Then we considered a far more complex three-dimensional model binary metallic glass, which was the motivating use case for the strategy.

In each case, to improve the computational tractability of our strategy, we partitioned the sampled LAEs into training and extension sets. Rather than calculating the GIIP distance between every pair of LAEs sampled, we calculated the GIIP distance only between pairs of members of the training set, and between members of the training and extension sets. This serves to reduce the number of GIIP distances calculated from $O(n_{total}^2)$ to $O(n_{train} \times n_{total})$, where $n_{total}$ and $n_{train}$ are the number of LAEs in the full dataset and the training dataset, respectively. We performed agglomerative clustering first solely on the training set to establish classes of LAEs, and then assigned LAEs in the extension set to the class to which the nearest LAE in the training set belonged. In a similar vein, we calculated the diffusion map first only on the LAEs in the training set to define a set of diffusion coordinates for the system, and then used the Nyström extension[31] to map the extension-set LAEs into those already-defined diffusion coordinates. This approach enabled us to include many more atoms in our analysis than computational limitations would permit otherwise.

The GIIP distance is constructed so that the distances between configurations have units of "atoms of difference." For example, if LAE $\mathcal{K}$ is identical to LAE $\mathcal{L}$ with the exception of $\mathcal{L}$ having a single void where $\mathcal{K}$ has an atom, the GIIP distance between $\mathcal{K}$ and $\mathcal{L}$ would be approximately one. In binary and higher-order systems, an atom with chemical species U aligning with an atom with chemical species V would produce a GIIP distance of approximately two. Minor atomic misalignments between otherwise similar LAEs would generally produce a GIIP distance between zero and one.

### Two-dimensional crystal

We modeled a defective two-dimensional crystal of 19,548 atoms with a Lennard-Jones interatomic potential. A portion of the sample

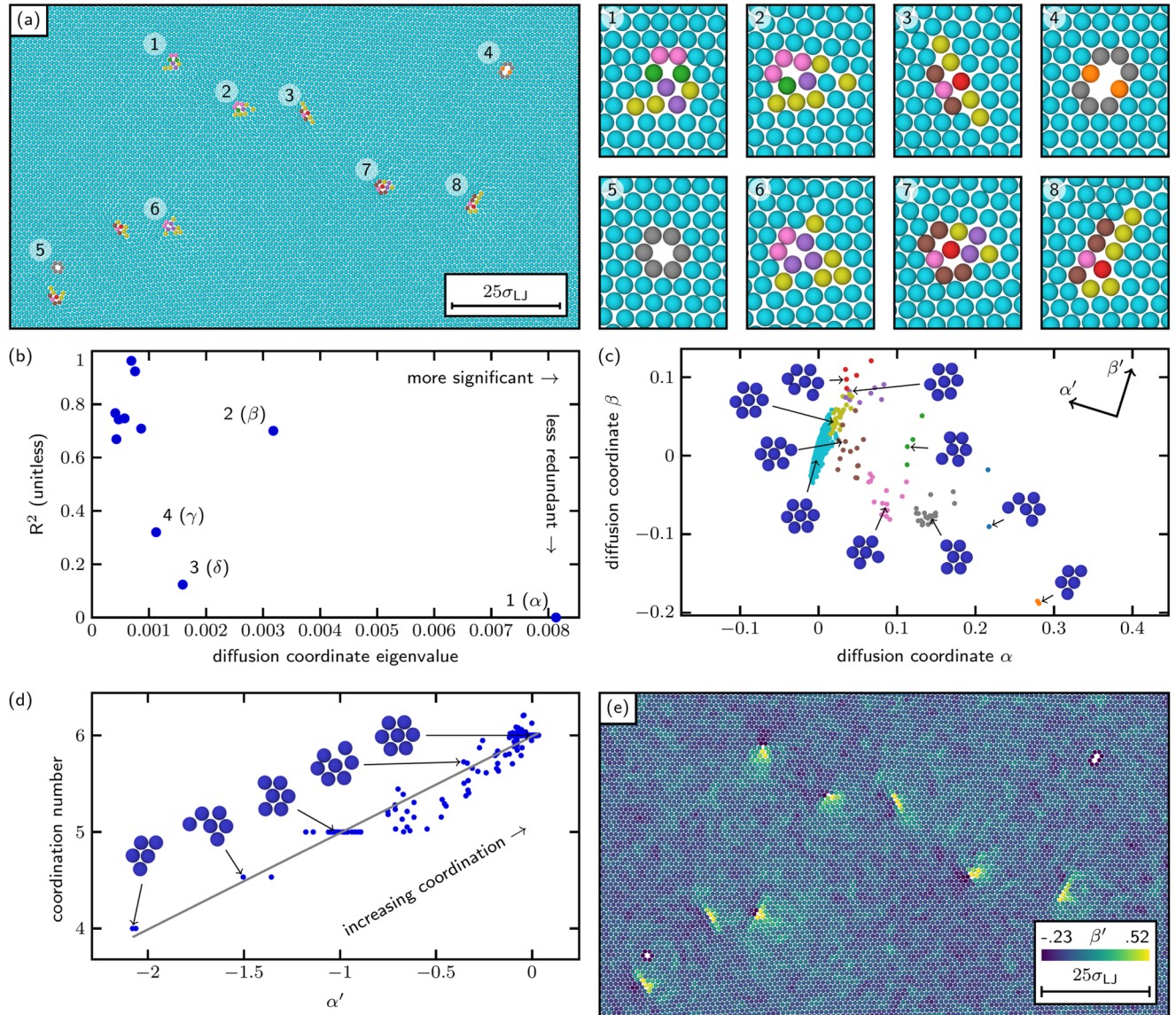

**Fig. 1 | Analysis of two-dimensional crystal. a** Rendering of two-dimensional crystal colored by membership in ten classes with length scales shown in terms of Lenard-Jones length scale $\sigma_{LJ}$. **b** Plot of the first ten diffusion coordinates in terms of significance and redundancy (captured by coefficient of multiple determination $R^2$), showing clear separation between the first four diffusion coordinates ($\alpha$, $\beta$, $\delta$, and $\gamma$) and the remainder. **c** The sample plotted in the $\alpha \times \beta$-plane of diffusion space, colored by class as in **a**, with exemplar environments shown. Directions $\alpha'$ and $\beta'$ in diffusion space are additionally noted. **d** $\alpha'$ correlates to coordination number with a coefficient of determination ($r^2$) of 0.93. Sample LAEs shown alongside. **e** The atoms from **a** colored by $\beta'$, revealing two-lobed volumetric strain fields as would be expected around dislocation cores. Source data are provided as a Source Data file.Source data

is shown in Fig. 1a. We identified 15,484 atoms away from the sample edges and ordered them in terms of potential energy. We selected a training set consisting of first-nearest-neighborhood LAEs centered on the 100 atoms with the lowest potential energy, and the 4900 atoms with the highest potential energy, and designated the remaining 15,484−5000 = 10,484 LAEs as the extension set. (We note that a hand-picked training set such as the one described above would invalidate or degrade many analyses, but can be acceptable for the specific task of parameterizing a nonlinear manifold. The goal of the training set is not to be statistically representative of the particular sampling of the manifold in the full dataset; rather, the goal is for the training set to thoroughly sample the shape of the manifold. Tight clusters of nearly identical samples—such low-energy crystalline configurations in this example—convey very little information about the manifold, whereas the higher-energy defective states are spread across the manifold in a very useful way.) We evaluated the GIIP distances between the sampled LAEs and, applying

agglomerative clustering to the raw GIIP distances, found that the LAEs could be partitioned into 10 classes with no two members of any class being more than one atom distant from each other (by GIIP distance). The atoms in Fig. 1a are colored according to membership in these 10 classes. We note at this point that in an agglomerative clustering scheme the number of classes are ultimately a design decision driven by the purpose of the classification project. For example, if the purpose of LAE classification is as a conceptual research aid, minimizing the number of classes would be of paramount importance (which would in turn drive tolerance of within-class variation). On the other hand, if the purpose of LAE classification is as the basis of an off-lattice kinetic Monte Carlo algorithm, classes would need to contain only LAEs with similar available kinetic pathways, necessitating far less tolerance of within-class variation, and consequently more classes. The trade-off between within-class variation and number of classes is visualized for this two-dimensional crystal example in the Supplementary Material.

We also generated the diffusion coordinates for the first-nearest-neighborhood LAEs in the sample, using the training and extension set approach described above. Each diffusion coordinate was associated with an eigenvalue that expresses the strength of the coordinate's contribution to variation within the data−that is, larger eigenvalues are more important, while smaller eigenvalues can be truncated without losing significant information. We further examined the redundancy of each successive diffusion coordinate (indexed by $i$), by attempting to predict it in terms of the preceding diffusion coordinates (1 through $i-1$) with polynomial regression. We calculated the coefficient of multiple determination ($R^2$) of the regression problem, where $R^2$ close to one indicates that the information in the diffusion coordinate of interest was already contained in the preceding diffusion coordinates, and $R^2$ closer to zero indicates that the diffusion coordinate contained new information (an analysis based on ref. 32). The eigenvalue and $R^2$ values for this diffusion map are shown in Fig. 1b, where the first four diffusion coordinates (labeled $\alpha$, $\beta$, $\delta$, and $\gamma$) are clearly separated. For this example, we take $\alpha$ through $\gamma$ to be our diffusion map.

In this simplified case it was clear that the first four diffusion coordinates are more valuable than the rest. However, for other materials, selecting the low-dimensional set of diffusion coordinates might be more arbitrary as redundancy and eigenvalue fade smoothly to 1 and 0, respectively. In these cases, thresholds for redundancy and importance would have to be selected that balance simplicity against detailed descriptiveness in view of the task at hand, in the same spirit as setting the number of classes.

The LAEs in our sample are plotted in the $(\alpha \times \beta)$-plane of diffusion space in Fig. 1c. The LAEs are colored by their 10 agglomerative clustering classes as in Fig. 1a, with exemplar LAEs rendered alongside for each class. The numeric values of the diffusion coordinates are entirely arbitrary, as they are simply a parameterization of a manifold, and could be shifted, scaled, or invertibly nonlinearly transformed and remain a valid parameterization of that manifold. The valuable information is in the relative positions of the points in diffusion space, how they cluster and the trends they reveal when the manifold is "unrolled" into simpler coordinates. The axes are notably not labeled in a physically intuitive way, as they only reflect trends in the variation of the data; it is our job to inspect, regress, and interpret the meanings of those trends.

On inspection of Fig. 1c we observe that the agglomerative clustering classes generated from the raw GIIP distance matrix are clustered in diffusion space. A tight grouping of near-perfect-crystal LAEs appear near $\alpha = 0$, $\beta = 0$ with defective LAEs scattered alongside. The coordination number of the central atom in each LAE varies consistently along the direction marked $\alpha'$ in Fig. 1c, a correlation confirmed in Fig. 1d (noting again the arbitrary nature of the units of $\alpha'$) by fitting a line with a coefficient of determination ($r^2$) of 0.93. Taking $\beta'$ to be orthogonal to $\alpha'$ in the $\alpha \times \beta$-plane, and using $\beta'$ to color the atoms in Fig. 1e reveals volumetric strain fields in the vicinity of dislocation cores in the sample, where negative $\beta'$ appears in connection with the compressive fields of extra half-planes of atoms, and positive $\beta'$ appears as balancing tensile fields mirrored across the slip plane[33].

In principle, the methods described in this section could readily be applied to a three-dimensional crystal to extract defects, but in our view there are already widely-used better-suited tools for this task. So, having illustrated our approach, rather than invest in further interpretation of a contrived example, we move on to a far more complicated and interesting system, namely a three-dimensional binary metallic glass.

## Three-dimensional binary metallic glass

We used an embedded atom method (EAM) interatomic potential to model a three-dimensional equicompositional NiNb metallic glass[34–36]. We generated four melt-quench samples of size 13500 atoms each, cooled (respectively) at rates of $1.7 \times 10^{13}$, $3.4 \times 10^{12}$, $1.7 \times 10^{12}$, and $1.7 \times 10^{11}$ K s$^{-1}$. A representative rendering of the sample cooled at $1.7 \times 10^{11}$ K s$^{-1}$ is shown in Fig. 2a. The dataset was constructed by pooling the first-nearest-neighborhood LAEs centered on Ni atoms from all four quench-rate samples (27,000 LAEs total). (We also ran a small-scale ($n = 3000$) analysis on Nb-centered LAEs, but focused our computational resources on Ni-centered LAEs when we found that more interesting structure emerged around Ni atoms. We withhold the results of our small-scale Nb-centered analysis due to space constraints and doubts over whether 3000 LAEs are representative of the glass.) The data was randomly partitioned into a training set consisting of 1700 LAEs from each quench-rate sample (6800 training LAEs total) an extension set consisting of the remaining 20,200 LAEs. As before, we evaluated GIIP distances (including species information, so Ni atoms do not align constructively with Nb atoms) and calculated the diffusion map for the sample. An initial attempt at agglomerative clustering found that for this metallic glass (MG) sample, the within-class variation remained high for any number of classes (illustrated in the supplementary material). In other words, we found little evidence to support the notion that the local structure of metallic glass is amenable to discrete dimensionality reduction. For the purpose of coloring our diffusion space scatter plots, we partitioned the data into twenty classes using agglomerative clustering but make no claim as to their physical significance.

The diffusion coordinates are plotted in Fig. 2b in terms of significance and redundancy, and from this we selected a parsimonious set of five diffusion coordinates (1, 2, 3, 5, and 6) that are well-separated from the other diffusion coordinates. These are labeled $\mathcal{A}$ through $\mathcal{E}$. Figure 2c shows that diffusion coordinate $\mathcal{A}$ correlates to a high degree with the chemical composition of the local environment.

The LAEs in the sample are plotted in the $(\mathcal{B} \times \mathcal{C})$-plane of diffusion space in Fig. 2d, colored by their membership in the twenty agglomerative clustering classes, with representative LAEs from the convex hull rendered around the perimeter. We find that the LAEs populate a roughly triangular region in this plane, with the corners representing three extremes of symmetry. At one corner (negative $\mathcal{B}$, small $\mathcal{C}$) we find LAEs with strong tetrahedral symmetry. At another corner (positive $\mathcal{B}$ and $\mathcal{C}$) we find LAEs tending towards cyclic ($C_n$) symmetry−meaning rotational symmetry about a single axis[37], but few other symmetries. At the last corner (negative $\mathcal{C}$ and small $\mathcal{B}$) we find LAEs with prismatic (particularly $D_{2h}$) symmetry[37].

The LAEs are also plotted in the $(\mathcal{A} \times \mathcal{D})$-plane in Fig. 2e, colored as in Fig. 2d. We have been unable to develop physical interpretations of $\mathcal{D}$ or $\mathcal{E}$, but note the presence of strong cusps in the envelope of LAEs sampled. These cusps suggest the existence of extrema, either energetic or symmetric, where the number of observed configurations drops to one, meaning that a single configuration is overwhelmingly favored at that point. Kernel density estimate plots of the $(\mathcal{B} \times \mathcal{C})$ and $(\mathcal{A} \times \mathcal{D})$ planes are provided in the Supplementary Material.

Next, we consider the impact of quench rate on the structural state of our model metallic glass.

## Effect of quench rate on metallic glass

It is well-established that quench rate impacts both the structure and properties of metallic glass[38], as slower quench rates allow greater numbers of LAEs to find their way to lower-energy states. We separated out the LAEs from our dataset that come from the fastest-quenched and slowest-quenched samples ($1.7 \times 10^{13}$ and $1.7 \times 10^{11}$ K s$^{-1}$, respectively) and compared partial radial distribution functions (Fig. 3a) and distributions in diffusion space (Fig. 3b, c). Subtle differences between the partial radial distribution functions are observed. Most prominently, the first peak of the Ni-Nb distributions is slightly increased for the slow quench relative to the fast quench, suggesting that slower quenches encourage Nb enrichment of Ni-centered LAEs. In Fig. 3b we examine the distributions of chemical compositions of the LAEs (equivalent to diffusion coordinate $\mathcal{A}$) and find that slower quench rates indeed

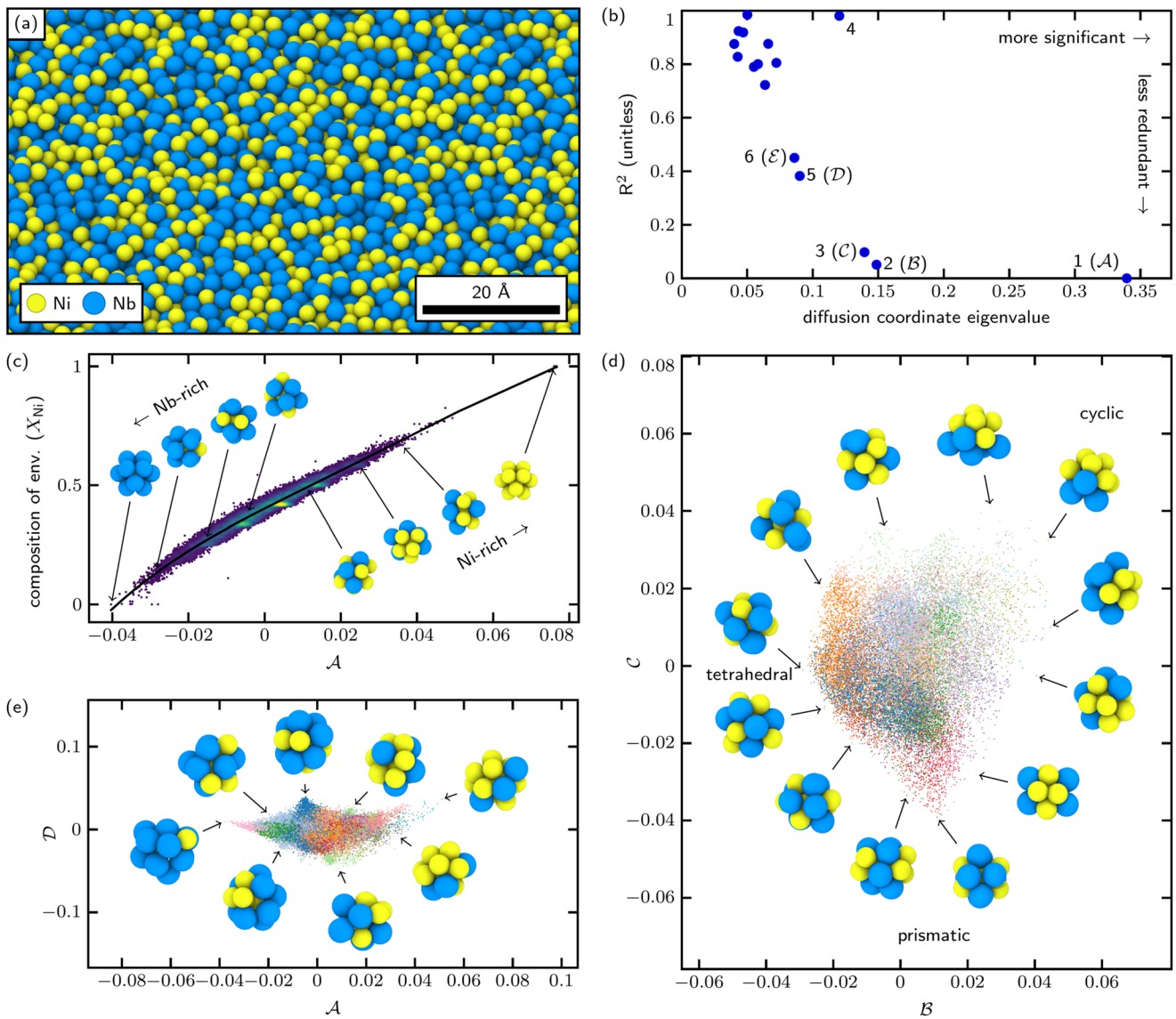

**Fig. 2 | Analysis of NiNb glass. a** Rendering of the NiNb glass. **b** Diffusion coordinates plotted in terms of significance and redundancy (captured by coefficient of multiple determination $R^2$), showing most significant diffusion coordinates $\mathcal{A}$, $\mathcal{B}$, $\mathcal{C}$, $\mathcal{D}$, and $\mathcal{E}$. **c** Diffusion coordinate $\mathcal{A}$ correlates to the composition of the first-nearest-neighbor environments ($X_{Ni}$) (coefficient of determination $r^2 = 0.97$), plotted in terms of the fraction of Ni neighbors in the LAE (excluding the central atom), with points colored by kernel density estimate. Example LAEs are shown alongside. **d** The sampled configurations plotted in the $\mathcal{B} \times \mathcal{C}$-plane of diffusion space, colored by membership in 20 agglomerative clustering classes, with representative configurations shown. The configurations fall in a roughly triangular space with one corner representing tetrahedral symmetry, one corner representing prismatic symmetry, and one corner representing cyclic configurations. Sample LAEs shown alongside. **e** Sampled configurations plotted in the $\mathcal{A} \times \mathcal{D}$-plane of diffusion space, colored by class as before. Cusps in the data are noted, suggesting that (for example) certain compositions strongly favor certain structures. Sample LAEs shown alongside. Source data are provided as a Source Data file.Source data

disfavor concentrated Ni-rich LAEs in favor of slightly Nb-rich LAEs, supporting and adding detail to the interpretation of the radial distribution functions. This implies a thermodynamic drive away from segregation as cooling rates increase, but is not particularly detailed in terms of the actual structural transitions occurring in the glass.

To further illuminate these changes, we plot the difference in probability density functions for the two quench rates in the ($\mathcal{B} \times \mathcal{C}$)-plane in Fig. 3c. In this plot, red indicates regions of diffusion space that are relatively denuded at the slow quench rate, while blue indicates enrichment in the slow sample. We find that slower cooling rates strongly disfavor LAEs with local cyclic symmetry and intermediate configurations, are close to neutral with respect to prismatic LAEs, and strongly favor LAEs with tetrahedral symmetry (additional quantitative details in the Supplementary Information). This is consistent with the previously-mentioned trend towards ordering of Ni

and Nb atoms, which are known to form intermetallic $Ni_3Nb$ and $Ni_6Nb_7$ phases at equilibrium[39] and illustrates one of the steps that take a sample from liquid to intermetallic states[40].

As a final note, in an effort to determine whether diffusion coordinates trained on individual quench datasets parameterized the same manifold, we calculated four sets of diffusion coordinates, with each set trained on a single quench but extended across all four quenches. We then used multivariable polynomial regression to analyze whether these four sets of diffusion coordinates were mutually bijective. We found that we were able to regress from each set of diffusion coordinates to the low-order terms of the other sets of diffusion coordinates with all adjusted $R^2$ values greater than 0.997. This strongly suggests that training separately on single quench rate datasets produces diffusion maps parameterizing the same manifold i.e. encoding the same information.

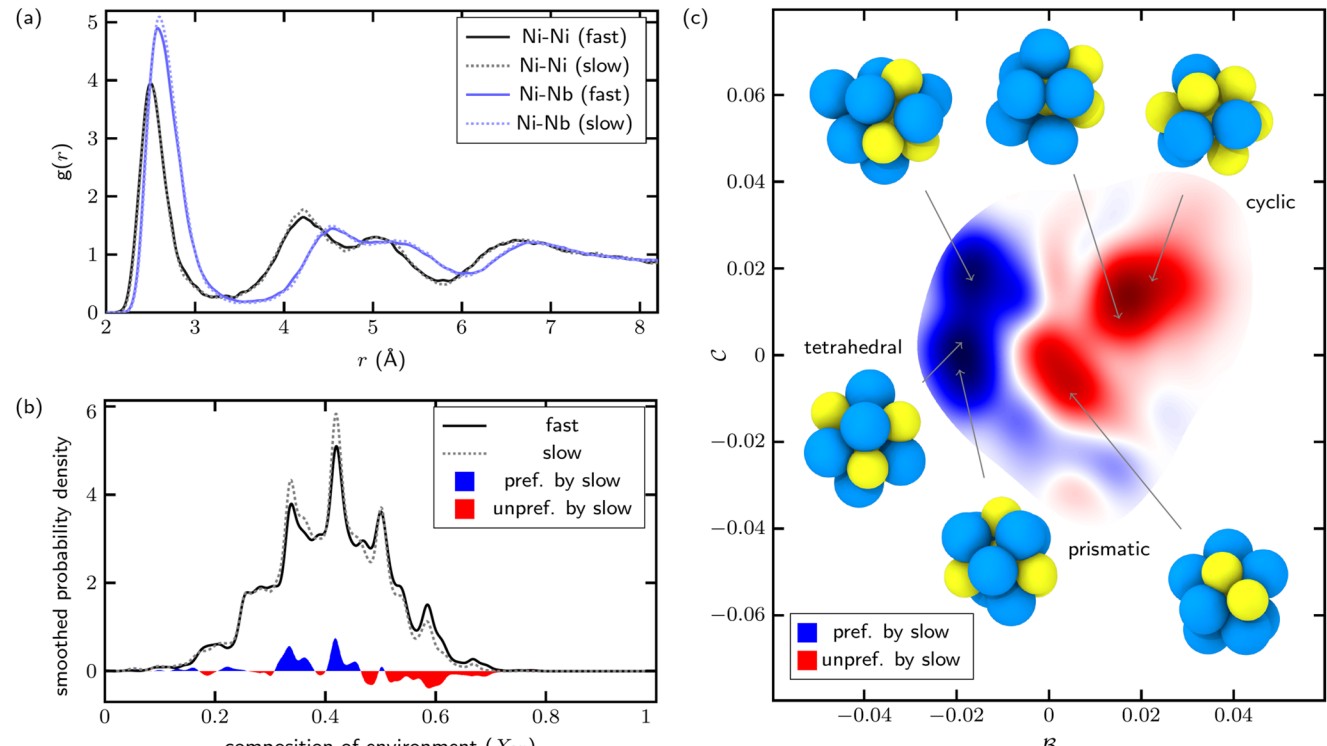

**Fig. 3 | Comparison of NiNb glass samples across quench rates in diffusion space, as defined in Fig. 2. a** Ni-centered partial radial distribution functions ($g(r)$) for the fastest-quenched and slowest-quenched samples in our metallic glass dataset ($1.7 \times 10^{13}$ and $1.7 \times 10^{11}$ K s$^{-1}$, respectively). **b** Smoothed probability density functions of local atomic environment (LAE) composition ($X_{Ni}$) for fastest and slowest quench rates, and difference in blue and red along the horizontal axis. Blue indicates increased density at the slower quench rate, while red indicates decreased density. **c** Difference in smoothed probability density functions for the fastest and slowest quench rates in the $\mathcal{B} \times \mathcal{C}$-plane of diffusion space; red indicates decreased density at the slower quench rate, while blue indicates higher density at the slower quench rate. Slower quench rates seem to favor tetrahedral configurations at the expense of cyclic configurations. Exemplar LAEs shown alongside. Source data are provided as a Source Data file.Source data

## Discussion

Local structural descriptors for glass currently in the literature generally involve a trade-off between simplicity (interpretability), and completeness (generalizability). Our approach emphasizes completeness/generalizability and uses data mining to extract trends in the LAEs encountered to serve as a sort of compatibility layer between raw atomic coordinates and a human mind. In the case of the three-dimensional metallic glass, most of the LAEs sampled had about 21 near neighbors within 4 angstroms of the central atom, implying the need for an 84-dimensional space to describe them. The combination of GIIP and diffusion coordinates reduces this to a five-dimensional space that captures the dominant trends in the first-nearest-neighbor structure of the glass. This supports the postulate that the atomic configurations of a metallic glass sit on a (relatively) low-dimensional manifold despite featuring tens of thousands of possible distinct configurations.

Some descriptors in the literature focus on interatomic potential development based on spherical and hyperspherical harmonic expansions (e.g. SOAP[41]). These descriptors are general, and comparison to GIIP in this particular use case is somewhat artificial, since they are highly optimized for a very different use case from structure identification. We also note that GIIP would be an inefficient method for interatomic potential development. It is worth noting that these descriptors systematically discard structural information (even when truncating their Fourier bases at arbitrarily high numbers of terms)[30], so distance measurements based on these descriptors are inherently lossy. The effects of this will vary on a case-by-case basis but an incomplete generalized distance function is at risk of "short-circuiting"[42] the dimensionality reduction methods presented here.

However, the speed with which these descriptors can be evaluated might in some cases make this risk worthwhile, particularly for a first-pass structural assessment. We evaluate generalized distances based on the radial distribution function along similar lines.

One potential use of learned descriptors is as predictors of the local kinetic pathways available to atoms[24]. To the extent that fine-grained atomic behavior follows from static atomic structure in MGs (see ref. [43]), our GIIP-based descriptors are a potential (if very slow) approach for reducing the dimensionality of the predictor domain. Stepping back, however, we note the dynamic nature of both short-range and medium-range order in metallic glasses at finite temperature[44,45], the chaotic nature of which may defeat the predictive power of static structural descriptors in general.

Descriptors based on diffusion maps and agglomerative clustering are also not lossless, even when the distance function satisfies the identity of indiscernibles. Our approach deals with this by being tunably lossy, letting the end user pick the number of discrete classes or diffusion coordinates depending on use case.

It is perhaps more fair to compare our approach to the Z-clusters concept for metallic glasses. In some sense a large set of agglomerative clusters extracted by our approach can be understood as a more granular extension of Z-clusters, from which Z-clusters could be extracted. The Z-clusters approach is inherently discrete in nature, whereas our diffusion maps return a continuous parameterization of structure. The latter has the potential to be more useful than a discrete parameterization in some settings, depending on success of efforts to interpret the diffusion coordinates.

Comparison of our approach to free or flexibility volume, as well as topological descriptors for silicate glass, highlights its greatest

weakness, namely that the computer does not label the coordinates or classes that it returns. Our analyses suggest that it can be possible to identify some coordinates through intuition and trial-and-error; mapping the remaining coordinates to existing descriptors is an ongoing effort, as is developing physical intuition for the coordinates that don't seem to line up with existing ideas. While the opacity of parameters is a current challenge for the application of data-mining to the physical sciences, detection of structural trends, even when difficult to interpret, is ultimately a useful step towards both property prediction and physical insight.

The shortcoming of only partial interpretability is offset by the limited number of user-supplied assumptions underpinning our approach. The GIIP distance simply defines configurations with similar atomic positions as similar and allows for the determination of trends in the data without bias toward or away from principles rooted in crystalline materials science.

There remain open questions around the degree to which the model NiNb metallic glass studied here is structurally similar to physical NiNb glass, and to which the structural insights observed here might extend to other compositions and systems. We look forward to connecting GIIP to a growing body of atomistic experimental data for metallic glass (for example, refs. 36,46); we also look forward to applying this methodology across a wide range of model metallic glass systems.

We note that local structure alone is inadequate to predict the behavior of metallic glasses[45,47]. Our approach does not directly deal with longer-range fluctuations in glassy structure[44] that play an important role in e.g. plasticity. However, we hope that improved descriptors of local structure will enhance the conversation around structural heterogeneity by enabling more precise descriptions of the nature of that heterogeneity. We do hypothesize that to some extent the properties of a material will be determined by the volume-averaged distributions of first-nearest-neighbor LAEs in diffusion space—something like a texture map for a polycrystalline metal—but note that spatial correlations between even fully-characterized LAEs remain difficult to engage. The examples presented in this paper are restricted to first-nearest-neighbor LAEs, but this is not a fundamental limitation of our strategy and it is possible that examination of second- and higher-order neighborhoods will prove illuminating.

In addition to ongoing work to make connections between the data-mined descriptors and existing physical descriptors, we anticipate that diffusion coordinates or their distributions can serve as state variables for constitutive modeling or as microstate definitions in statistical mechanical calculations. A logical next step in this approach is also to data-mine individual kinetic transitions in the glass. Finally, we note that our approach is material-agnostic and has potential applications in the study of liquids, polymers, colloidal systems, granular systems, and other non-crystalline materials. The rational design of materials requires a fundamental understanding of structure. This methodology is a step towards illuminating glassy structure in a form that is detailed yet interpretable, and a potential step towards a universal coordinate system of machine-learned structural descriptors parameterizing the manifold of physically realizable materials.

## Methods

### Gaussian Integral Inner Product (GIIP) distance

The Gaussian Integral Inner Product is inspired by the Smooth Overlap of Atomic Positions (SOAP) formulation[41], centering a Gaussian function on each atom to create a continuous atomic density function, and then comparing atomic density functions as proxies for their corresponding (discrete) atomic environments. The various concepts in this section are illustrated in simplified form in Fig. 4.

We preliminarily establish the integral inner product of two functions in three-dimensional real space and the norm induced by that integral inner product (where $a$ and $b$ are arbitrary functions such

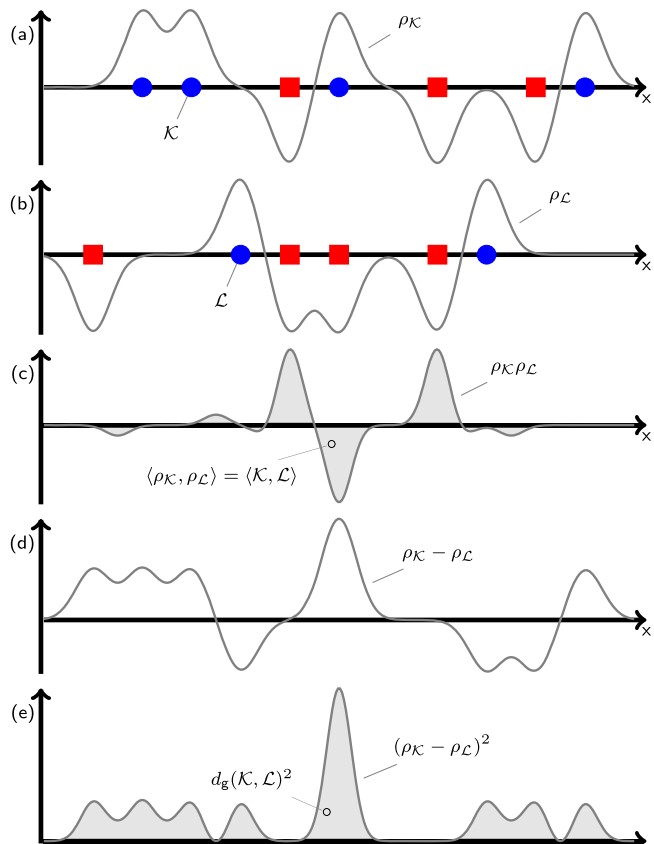

**Fig. 4 | One-dimensional schematic of the Gaussian Integral Inner Product and induced distance.** Subfigures **a** and **b** respectively show LAEs $\mathcal{K}$ and $\mathcal{L}$, each of which contain atoms of two types (indicated by blue circles and red squares) in space ($x$), and the atomic density functions $\rho_\mathcal{K}$ and $\rho_\mathcal{L}$ formed by placing weighted Gaussians at each atom site. This schematic illustrates the sign-species convention for Gaussian weights, where one atom type has positive weights, and the other atom type has negative weights. Subfigure **c** shows the product of the atomic density functions $\rho_\mathcal{K}$ and $\rho_\mathcal{L}$, with the integrated shaded area being equal to the Gaussian Integral Inner Product $\langle\mathcal{K},\mathcal{L}\rangle$. Subfigure **d** shows the difference between the $\rho_\mathcal{K}$ and $\rho_\mathcal{L}$, the square of which is shown in **e**. The integrated shaded area in **e** is equal to the Gaussian Integral Inner Product distance $d_g(\mathcal{K},\mathcal{L})^2$.

that the integrals below are convergent)[48].

$$\langle a,b \rangle = \int_{\mathbb{R}^3} a(\boldsymbol{x}') \cdot b(\boldsymbol{x}') \, d\boldsymbol{x}' \tag{1}$$

$$||a|| = \sqrt{\langle a,a \rangle} \tag{2}$$

We construct a three-dimensional Gaussian function $G_s$ centered at zero with width $s$ and normalized so that the integral inner product norm (Eq. (2)) of $G_s$ is unity.

$$G_s(\boldsymbol{x}) = \exp\left[-|\boldsymbol{x}|^2/(2s^2)\right]/(\pi^{3/4}s^{3/2}) \tag{3}$$

We will use $G$ to construct atomic density functions from atomic configurations, illustrated in Fig. 4a, b for a simple one-dimensional case. Let $\mathcal{K}$ be an LAE, consisting of a set of vectors pointing to atomic positions, and a corresponding set of species for each atom; while these could be stored as a matrix of positions and a vector of species, we note that the atoms in an LAE are unordered. We place no restrictions on the size or nature of the LAE, which could be constructed by selecting a particular number of nearest neighbors, or by setting a cutoff radius. The atomic positions might optionally be

shifted to center on a focal atomic position. We define an atomic density function $\rho_{\mathcal{K}}$ by placing a weighted, shifted $G$ gaussian function at each atomic position in $\mathcal{K}$. We will describe using the weights ($w$) to encode species information and to handle cutoff radius behavior at a later point in this section. Note that in addition to each atom having its own weight, each atom can also have its own Gaussian width $s$, though in practice we generally use a constant $s$ for all atoms in a configuration. We write $\rho_{\mathcal{K}}$ in terms of a sum over atoms $\mathbf{y} \in \mathcal{K}$, where $\mathbf{y}$ acts as both an index for weights and Gaussian widths, and as a position vector.

$$\rho_{\mathcal{K}}(\mathbf{x}) = \sum_{\mathbf{y} \in \mathcal{K}} w_{\mathbf{y}} G_{s_{\mathbf{y}}}(\mathbf{x} - \mathbf{y}) \tag{4}$$

Combining Eqs. (1) and (4), we define the Gaussian Integral Inner Product (GIIP) between two LAEs ($\mathcal{K}$ and $\mathcal{L}$):

$$\langle \mathcal{K}, \mathcal{L} \rangle = \langle \rho_{\mathcal{K}}, \rho_{\mathcal{L}} \rangle \tag{5}$$

illustrated schematically in Fig. 4c. We can also define the GIIP distance between those LAEs using Eq. (2) and the distributive property of the integral inner product:

$$d_{\mathrm{g}}(\mathcal{K}, \mathcal{L})^2 = ||\rho_{\mathcal{K}} - \rho_{\mathcal{L}}||^2 \tag{6}$$

$$= \langle \rho_{\mathcal{K}} - \rho_{\mathcal{L}}, \rho_{\mathcal{K}} - \rho_{\mathcal{L}} \rangle \tag{7}$$

$$= \langle \rho_{\mathcal{K}}, \rho_{\mathcal{K}} \rangle + \langle \rho_{\mathcal{L}}, \rho_{\mathcal{L}} \rangle - 2 \cdot \langle \rho_{\mathcal{K}}, \rho_{\mathcal{L}} \rangle \tag{8}$$

$$= \langle \mathcal{K}, \mathcal{K} \rangle + \langle \mathcal{L}, \mathcal{L} \rangle - 2 \cdot \langle \mathcal{K}, \mathcal{L} \rangle \tag{9}$$

illustrated in Fig. 4d, e. The GIIP distance defined above is immediately usable as an orientation-sensitive measurement between atomic configurations. It is possible to render the GIIP distance orientation-invariant by finding the minimum distance over all possible orientations:

$$d_{\mathrm{g,inv}}(\mathcal{K}, \mathcal{L})^2 = \min_{\mathbf{R} \in O(3)} d_{\mathrm{g}}(\mathcal{K}, \mathbf{R}\mathcal{L})^2 \tag{10}$$

where $O(3)$ is the three-dimensional orthogonal group (comprising rotations and rotoinversions) and $\mathbf{R}\mathcal{L}$ is a specific rigid-body rotation or rotoinversion of $\mathcal{L}$; note that in cases where chirality must be preserved, this minimization can be done over $SO(3)$ instead.

The atomic weights in the GIIP formulation serve three purposes: to exclude atoms from analysis (e.g. outside of a cutoff radius), to enforce continuity or smoothness with respect to perturbation of atomic positions near (for example) a cutoff radius, and to encode species information. Atoms may be excluded from analysis by setting their weights to zero, eliminating their effect on the atomic density function (Eq. (4)). This might be desirable when excluding extraneous information (perhaps, for example, ignoring the positions of hydrogen atoms in a polymer), or when excluding regions from analysis, as in the case of atoms outside of a cutoff radius. In cases where continuity or smoothness at a cutoff radius is desired, weights can be thought of as being the product of a function of the norm of atomic position and of a function of species (where, again, $\mathbf{y}$ acts as both an index for species and as an atomic position):

$$w_{\mathbf{y}} = f(|\mathbf{y}|)g(\mathrm{species}(\mathbf{y})) \tag{11}$$

By having $f$ continuously or smoothly vanish as it approaches the cutoff radius from below, continuity and smoothness of the GIIP distance with respect to atomic position are enforced across that cutoff radius. We suggest two strategies for encoding species information into weights. The first strategy, which we refer to as "sign-species," is

effective for binary systems, and consists of setting weights corresponding to one atom type to positive values, and weights corresponding to the second atom type to negative values. The second strategy, which we refer to as "vector-species," is effective for binary and higher-order systems and consists of setting weights for an $n$-ary system to be $n$-vectors, where the weight vector for an atom of species U will consist entirely of zeroes except for its coordinate corresponding to species U. This approach essentially reduces to computing $n$ separate GIIP distances, each excluding all but a single atom type, and then summing to obtain a single-valued distance value. Note that this summation must occur inside the optimization loop when computing the orientation-invariant GIIP distance in Eq. (10).

The GIIP formalism is equally applicable in one, two, or three dimensions and analytically tractable by substituting Eqs. (3) and (4) into (5) and integrating (1):

$$\langle \mathcal{K}, \mathcal{L} \rangle = 2\sqrt{2} \sum_{\mathbf{y} \in \mathcal{K}} \sum_{\mathbf{z} \in \mathcal{L}} w_{\mathbf{y}} w_{\mathbf{z}} \left( \frac{s_{\mathbf{y}} s_{\mathbf{z}}}{s_{\mathbf{y}}^2 + s_{\mathbf{z}}^2} \right)^{3/2} \exp\left[ -|\mathbf{y} - \mathbf{z}|^2 / (2s_{\mathbf{y}}^2 + 2s_{\mathbf{z}}^2) \right] \tag{12}$$

which, in the case where $s$ is constant for all atoms, simplifies to:

$$\langle \mathcal{K}, \mathcal{L} \rangle = \sum_{\mathbf{y} \in \mathcal{K}} \sum_{\mathbf{z} \in \mathcal{L}} w_{\mathbf{y}} w_{\mathbf{z}} \exp\left[ -|\mathbf{y} - \mathbf{z}|^2 / (4s^2) \right] \tag{13}$$

We implemented Eqs. (9), (10), (11), and (13) in a Python library based on the popular PyTorch library[49] with both thread-based and GPU-based parallelism, and executed the computations shown here on computers ranging from a laptop to a large institutional cluster. The computations were accelerated by condensing all atomic positions into tensors of size $n_{\mathrm{configurations}} \times \max\{n_{\mathrm{atoms}}\} \times n_{\mathrm{dimensions}}$, and all weights into tensors of size $n_{\mathrm{configurations}} \times \max\{n_{\mathrm{atoms}}\}$ (where $\max\{n_{\mathrm{atoms}}\}$ represents the maximum number of atoms found in any configuration). In neighborhoods with fewer atoms than $\max\{n_{\mathrm{atoms}}\}$, weights corresponding to non-existent atoms are set to zero. We used the hyperspherical-coverings library[50] to sample orientation space in evaluating Eq. (10).

In the analysis of the unary two-dimensional crystal shown in Fig. 1, we used the sum of the weights in an LAE as the coordination number of the central atom of the LAE. For the three-dimensional NiNb glass, we used the sign-species formalism, so the sum of weights in the LAE was linearly related to the composition of the nearest neighbors of the central atom.

For the two-dimensional crystal, we used a uniform weight of 1 for atoms less than $1.3\sigma_{\mathrm{LJ}}$ (where $\sigma_{\mathrm{LJ}}$ is the length-scale of the Lennard-Jones formalism) from the central atom, and had the weight smoothly drop to zero (using a cubic spline) for atoms between $1.3\sigma_{\mathrm{LJ}}$ and $1.75\sigma_{\mathrm{LJ}}$ from the central atom; in our GIIP calculations we used $s = 0.5\sigma_{\mathrm{LJ}}$ and sampled orientation space with a resolution of 1 degree.

For the three-dimensional metallic glass we used a uniform weight of $+1/-1$ for Ni/Nb less than 3 Å from the central atom, and had the weight smoothly drop to zero between 3 Å and 4 Å. In our GIIP calculations we used $s = 1.0$ and sampled three-dimensional orientation space first with a coarse resolution of 5 degrees and then fine resolution of 1 degree around the minimum of the coarse search.

## Cluster analysis

Agglomerative clustering assigns datapoints to a prescribed number of classes, where each class contains datapoints that are similar by some measure. Here it enables us to break the atomic configurations into classes; for example, in a crystalline material we might establish classes as: a) atoms in a perfect lattice, b) atoms bordering a single vacancy, c) interstitial atoms, and so forth. In other words, agglomerative clustering provides a discrete parameterization of local structure. There

might be some variation among the configurations in each class (for example, due to thermal vibrations or strain fields), but for the classification to be useful the configurations within each class must be similar enough to behave similarly, with the threshold of similarity ultimately being a user decision. There are many potential agglomerative clustering algorithms available in literature; in this work we used a patient diameter-minimizing criterion[22] but it is possible that other clustering algorithms would yield superior results. We refer the reader to any textbook on data mining for conceptual details, and to the scipy.cluster package[51] for an accessible implementation.

## Diffusion maps

Dimensionality reduction algorithms are used for computing a set of data-driven latent coordinates. Here, we used Diffusion Maps[23], a manifold learning scheme. Diffusion maps offer a reparametrization of the original data by revealing its intrinsic geometry. Below we give a short description of the algorithm.

The diffusion maps algorithm is applied to a given data set $\mathbf{T} = \{t_i\}_i^n$ sampled from a manifold $\mathcal{M}$, where we assume $t_i \in \mathbb{R}^m$ first constructs a random walk on the data. This random walk is estimated based on the local similarity of the sampled data points. The similarity measure is computed in terms of a kernel, for example the Gaussian kernel, defined as

$$W(t_i, t_j) = \exp\left(\frac{-d\left(t_i - t_j\right)^2}{2\varepsilon^2}\right). \tag{14}$$

where $d(\cdot)$ is an appropriate norm; in our case the GIIP distance. The hyper-parameter $\varepsilon$ is the kernel's scale. In our work we selected $\varepsilon$ by trial-and-error; for the two-dimensional crystal we used $\varepsilon = 1.0$ atoms of GIIP distance and for the three-dimensional metallic glass we used $\varepsilon = 1.5$ atoms of GIIP distance.

To recover a parametrization of the data set regardless of its sampling density a normalization on $\mathbf{W}$ is computed,

$$\widetilde{\mathbf{W}} = \mathbf{P}^{-1}\mathbf{W}\mathbf{P}^{-1}. \tag{15}$$

where the diagonal matrix $\mathbf{P} \in \mathbb{R}^{n \times n}$ is computed by,

$$P_{ii} = \sum_{j=1}^{n} W_{jj} \tag{16}$$

A second normalization is then applied to $\widetilde{\mathbf{W}}$ to construct a Markovian matrix $\mathbf{M}$,

$$M(t_i, t_j) = \frac{\widetilde{W}(t_i, t_j)}{\sum_{j=1}^{n} \widetilde{W}(t_i, t_j)} \tag{17}$$

Computing the eigendecomposition of $\mathbf{M}$ gives a set of eigenvalues $\lambda_i$ and eigenvectors $\phi_i$

$$\mathbf{M}\phi_i = \lambda_i \phi_i \tag{18}$$

The eigenvectors of $\mathbf{M}$ are the data-driven coordinates that offer a reparametrization of $\mathbf{T}$. However, a selection of the eigenvectors that span independent directions is needed. Those independent eigenvectors are called non-harmonics, and we refer the reader to ref. 32 and also in the Supplementary Information of ref. 52 for a more detailed discussion of non-harmonic eigenvectors. If the number of those independent/non-harmonic eigenvectors is less than original dimensions of $\mathbf{T}$ then we claim the diffusion maps algorithm achieved dimensionality reduction.

To obtain the diffusion coordinates for points $t_{new} \notin \mathbf{T}$ without recomputing the entire diffusion map the Nyström Extension[31,53] can

be used. Nyström Extension computes the new coordinates based on a weighted average,

$$\phi_i(t_{new}) = \frac{1}{\lambda_i}\sum_{j=1}^{n} \widetilde{M}(t_{new}, t_j)\phi_i(t_j), \tag{19}$$

where $\phi_i(t_j)$ denotes the $j$-th component of the $i$-th eigenvector and $\phi_i(t_{new})$ is the estimated $i$-coordinate for the out of sample data point $t_{new}$. To compute the kernel $\widetilde{M}$ the same normalizations and the same scale parameter $\varepsilon$ used for diffusion maps are needed.

Finally, we note that the orientation-invariant GIIP distance does not satisfy the triangle inequality, with the consequence that the kernel matrix is only approximately symmetric positive definite. In our metallic glass dataset, the first hundred eigenvalues of the kernel matrix were positive, but several relatively small (roughly 1/2 of a percent of the largest eigenvalue in magnitude) negative eigenvalues appeared at higher orders. This places use of the GIIP distance outside of portions of theory for diffusion maps, but we deem our kernel matrix to be "close enough" to symmetric positive definiteness, and our results to be sensical enough to accept this, particularly in view of the absurdity of using more than one hundred diffusion coordinates in this dimensionality reduction exercise.

## Sample construction

The two-dimensional crystal was generated using the LAMMPS molecular dynamics package[54]. We generated 20,000 atoms arranged in a hexagonal lattice with vacuum boundaries, interacting with a standard 12-6 Lennard-Jones interatomic potential with unit mass, distance, and energy terms, and a cutoff at $2.5\sigma_{LJ}$, where $\sigma_{LJ}$ is the Lennard-Jones length scale. We then introduced defects into the crystal by randomly selecting an atom and one of its six close-packed nearest neighbors, and then deleting the half-plane of atoms in the direction of that neighbor. This was repeated ten times. After thus modifying the sample, the system was equilibrated in the NVE ensemble with a timestep of 0.003 for 30,000 time steps. This resulted in a system with roughly ten defects consisting of both dislocations and vacancies.

The four three-dimensional metallic glass samples were also generated using LAMMPS using an embedded atom method potential[55] constructed for NiNb glasses. For each sample, we generated 13,500 atoms in an FCC lattice (initial $a = 4$ Å), randomly assigning approximately half to be Ni and half to be Nb. We set the initial velocity of the atoms consistent with a temperature of 2500 K and then equilibrated the simulation in the NPT ensemble at a temperature of 2000 K and a pressure of 1 bar, with a timestep of 0.001 ps for 10,000 steps. Finally, we quenched the simulation in NPT mode from 2000 K to 300 K at quench rates for the four samples were $1.7 \times 10^{13}$, $3.4 \times 10^{12}$, $1.7 \times 10^{12}$, and $1.7 \times 10^{11}$ K s$^{-1}$, resulting in a glassy structure. These quench rates are too high to be experimentally relevant; however, high quench rates encouraged formation of a wide range of energetically unfavorable configurations for the manifold learning algorithm to learn.

## Data availability

The atomistic data associated with this manuscript are available at https://github.com/sandialabs/giip[56]; processed data underpinning figures in this paper are provided in the Source Data file accompanying this manuscript. Source data are provided with this paper.

## Code availability

The GIIP codebase is available at https://github.com/sandialabs/giip[56].

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

## Acknowledgements

Thanks to Ian Winter, Mark Wilson, and N. Scott Bobbitt for enlightening conversations. This work was funded by the President Harry S. Truman Fellowship in National Security Science and Engineering and Laboratory Directed Research and Development programs at Sandia National Laboratories. The JHU authors gratefully acknowledge support by subcontract under PO#2221045 from Sandia National Laboratories. This article has been authored by an employee of National Technology & Engineering Solutions of Sandia, LLC under Contract No. DE-NA0003525 with the U.S. Department of Energy (DOE). The employee owns all right, title and interest in and to the article and is solely responsible for its contents. The United States Government retains and the publisher, by accepting the article for publication, acknowledges that the United States Government retains a non-exclusive, paid-up, irrevocable, world-wide license to publish or reproduce the published form of this article or allow others to do so, for United States Government purposes. The DOE will provide public access to these results of federally sponsored research in accordance with the DOE Public Access Plan https://www.energy.gov/downloads/doe-public-access-plan. This paper describes objective technical results and analysis. Any subjective views or opinions that might be expressed in the paper do not necessarily represent the views of the U.S. Department of Energy or the United States Government.

## Author contributions

T.J.H. conceived of and implemented the GIIP formalism, prepared the manuscript, and obtained funding for this research. M.C. and S.F. provided atomistic data. R.M. and D.G. performed experiments testing GIIP and competing generalized distance functions. I.K. provided insights and expertise on diffusion maps, M.L.F. provided insights and expertise on the physics of metallic glasses, and M.D.S. conceived using a generalized distance function with diffusion maps to obtain the low-dimensional manifold for an atomistic system.

## Competing interests

The authors declare no competing interests.
