## [Peer Review File · Nature Communications]

Revealing the Hidden Structure of Disordered Materials by Parameterizing their Local Structural ManifoldREVIEWER COMMENTS

Reviewer #1 (Remarks to the Author):

The article may contain useful information. However, the manuscript is very difficult to read due to the vague definition of concepts.

The meaning of Fig. 1 and 2 cannot be understood from the previous text. Many variables used in text and graphics are not defined above. It is necessary to give a clear definition of all parameters used in the text before actual using them. What is r , x , α , α' , β , A , B , C , D , X ? And what is their physical meaning?

Some things become clear from the experimental procedure, but the meaning of many of the quantities used is not understood yet. What specific functions are used as "a and b"?

The authors state: "Let X be an atomic configuration, consisting of a set of vectors pointing to atomic positions." How many vectors are used? Where are they pointing? To the nearest atomic neighbors? How is it determined how many vectors are needed for each cluster? How are they recorded? In the form of a matrix?

What is plotted along the abscissa x -axis in Fig. 4?

Any work in the field of physics other than pure mathematics must have a clear physical meaning. In view of the above, I propose to make the necessary changes to the manuscript. Without a clear description of the variables, vectors and functions, further consideration of the work is impossible. Moreover, as a summary the authors state: "Applying this approach to a two-dimensional model crystal and a three-dimensional binary model metallic glass results in parameters interpretable as (i.e. one-to-one with) coordination number, composition, volumetric strain, and local symmetry. In particular, we show that a more slowly quenched glass has a higher degree of local tetrahedral symmetry at the expense of cyclic symmetry". Coordination number, composition, volumetric strain, and local symmetry can be easily obtained from the results of MD modeling by the existing methods. The fact that slowly quenched glass has a higher degree of local tetrahedral symmetry at the expense of cyclic symmetry may be also easily confirmed by the existing methods. What is the prime novelty of the proposed approach then? Please, state it clearly.

Reviewer #2 (Remarks to the Author):

Comments on "Quantifying the Structure of Disordered Materials" by Hardin et al.

Understanding the microstructure and its evolution in disordered materials is one of long-standing yet the most interesting problems in the field of condensed matter physics. Despite the various models proposed previously in the glass literatures, complete and precise characterization of its atomic-scale microstructure still faces a compelling challenge, particularly for metallic glasses (MGs). Unlike in crystalline materials, it is impossible to extract a unit cell to well describe the microstructure and particularly the structural defects of MGs owing to their lacking of long-range order.

The present manuscript reports on a strategy based on data mining to characterize the microstructural features of amorphous solid materials such as a binary Ni-Nb MG. The outcome of this work paves a useful and effective way to resolve the problem of trade-off between general applicability and interpretability in quantitatively characterizing its atomic-scale structure. Specifically, it seems to be successful in unraveling the atomic short-range ordering and its variation with cooling rate. This part is possibly compelling for some of researchers in the field of MG materials. However, this cannot lead to well establish the relationship between properties and microstructure for glassy solid materials. For instance, while the common wisdom suggests that MGs are homogeneous, a large number of experimental and theoretical investigations indicate that they are heterogeneous particularly in a dynamic sense. Therefore, the method proposed in the work in deepening the understanding of the microstructure and properties of MGs is limited. In this case, the reviewer does not recommend it to be published in Nature Communications.

Here are a few questions that need to be addressed/clarified.

1. For the two-dimensional crystal, the LAEs could be partitioned into 13 classes with the use of agglomerative clustering. However, for the metallic glass system, the authors claimed "An initial attempt at agglomerative clustering found that the variance in LAEs in the sample was so large...their physical significance". How did the authors partition the data of the three-dimensional glassy alloy? Are there any rules behind the classification?
2. The authors found that the LAEs plotted in (BCC)-plane of diffusion space (Fig. 2d) locate in a roughly triangular space, with three corners representing tetrahedral, cyclic, and prismatic symmetries, respectively. What are other planes of diffusion space look like? are there similar corners representing the symmetry information of the sample in these planes? If not, why only the (BCC)-plane of diffusion space?
3. Are the parameters found in this work, e.g., the one that could be interpreted as local symmetry, applicable to other NiNb alloys or other alloy systems?

Reviewer #3 (Remarks to the Author):

I apologize in advance to the authors for the length of this review. I tried to rigorously highlight the strengths and weaknesses of this work, while providing references, directions to consider, as well as optional requirements that I think would make this work more solid.

In this work, the authors presented a new metric, the Gaussian Integral Inner Product (GIIP) distance, that is used to directly assess the similarity or dissimilarity of a pair of local atomic environments (LAEs). An LAE is defined in terms of a smooth atomic density function (i.e. a weighted superposition of Gaussian functions) inspired by the SOAP formalism, which makes it robust to thermal fluctuations and to the definition of nearest neighbors. The GIIP distances between a large number of LAEs can then be used to partition them into a certain number of classes (clusters) by using an agglomerative clustering algorithm. These clusters are composed of similar LAEs, ideally leading to an interpretable partitioning of the system based on its structure. In addition to this partitioning, the GIIP distance can also be used to learn a manifold embedded in a high-dimensional space onto which the LAEs are distributed, using diffusion maps, a non-linear dimensionality reduction method. Being defined by a reduced number of latent variables (called "diffusion coordinates"), this manifold provides a lower-dimensional description of the structure of the system. Some of these diffusion coordinates are highly correlated to standard and well-known observables and order parameters, which provides a useful description of the heterogeneity of the system by using a limited number of variables.

This approach is first illustrated using an interpretable two-dimensional crystalline benchmark, for which the identified clusters are easily identifiable (i.e. crystal, vacancy, etc.) and some of the diffusion coordinates strongly correlated to known observables (coordination and volumetric strain).

After this useful proof of concept, the authors apply their method to a more challenging system: a three-dimensional binary metallic glass. Even though the results are now more difficult to interpret, as one can expect, they still present some interesting features: the main diffusion coordinate strongly correlates with chemical composition, and the distributions of the clusters in the reduced spaces formed by pairs of diffusion coordinates are strongly correlated to different symmetries of the LAE (cyclic, prismatic, etc.).

Before moving on to a more technical discussion about the methods and the results, I would first like to briefly summarize my general impression of this work: I find the manuscript well-written, clear, and organized in a pleasant way. In particular, I really appreciate the crystalline benchmark, which really helps understanding the approach with clear visual explanations, as it makes the understanding of the second analysis easier. The GIIP distance is a well-defined, elegant and easily interpretable metric to compare LAEs: expressing the difference between two LAEs in units of

"atoms of difference" is very intuitive. Finally, as will be discussed later in the review, several previous works have used dimensionality reduction methods and/or clustering algorithms in similar contexts. However, to the best of my knowledge, diffusion maps is a method that has not yet been used to study the local structure of glasses. I think that this is an interesting and powerful method that should be explored further in future work. And to conclude on the method itself: even though the results of this approach require post-hoc interpretation, this should obviously be contrasted to the commonly called "black-box" nature of many machine-machine learning methods employed in some other works. The results of this study stay reasonably interpretable, with minimal effort, which is what machine learning methods should aim for in my opinion.

Concerning the novelty of this work: while the method itself is new and interesting, I do not find the approach particularly original unfortunately. Reasonably interpretable descriptors (to be contrasted with e.g. SOAP), dimensionality reduction methods and/or clustering algorithms have been used in the past in similar contexts (see e.g. 10.1021/acs.jpcb.6b02144, 10.1038/s41467-020-19286-8, 10.1063/1.5118867, 10.1063/5.0004732, 10.1063/5.0128265) [N.B. in an effort to be fully transparent: this includes some of my recent work, but this does not impact my review in any way, as discussed with the editor]. The general conclusions of the most recent of these works are also similar: latent variables generally correlate well with well-known observables, and clusters can sometimes provide a good insight on the variability of local structures, but they can also be difficult to interpret (especially if there is a large number of them). While there are obvious differences, there are also obvious similarities, and I believe that this study is unfortunately not contextualized enough, as the authors might not know about these previous works. They should try to recontextualize the methods and results, stress the differences, and emphasize on the strengths and weaknesses before it is suitable for publication. There are also some comments and questions that I would like the authors to address, as I believe it would help clarifying some choices and make some results more robust:

(1.a) How does diffusion maps compare to basic PCA? Manifold learning can be very useful when the high-dimensional distribution to study is complex and intricate, but in the case of disordered systems like glasses, one can expect a reasonably homogeneous and unimodal distribution (see 10.1063/5.0128265). In such a case, complex dimensionality reduction methods like autoencoders do not seem to perform particularly better than standard PCA, since there is no complex manifold to learn (i.e. only a simple projection from a high-dimensional space onto a lower-dimensional hyperplane).

(1.b) How does diffusion maps compare to other non-linear dimensionality reduction methods, that can also learn a complex manifold to project new samples onto it (e.g. neural network autoencoders, locally linear embedding, etc.)?

(2) I would appreciate a more detailed explanation on how the training sets are built. For instance, for the crystalline benchmark, only atoms at the two extremes of the potential energy distribution $p(U)$ are selected. Consequently, the training set is not statistically representative of the whole dataset since its elements are "hand-picked", which is generally not an intuitive choice when building a training set. Could the authors explain the motivation behind this choice?

(3) I found the "significance vs. redundancy" approach to select the diffusion coordinates very interesting. It is more interpretable and easier to justify than, e.g. a rule of thumb on the amount of explained variance with PCA. However, how is the final number of diffusion coordinates chosen? Is there a well-defined criterion or just a visual estimation based on this scatter plot?

(4) In Fig. 1(c), having to "redefine"/"realign" the latent variables into α' and β' so they better correlate with interpretable quantities (coordination and strain) seems a bit artificial. One should not have to "tweak" the latent variables: they are learned in an unsupervised way in the first place for that reason, so they can be directly used to learn about the structure. Also, α' and β' do not appear to be orthogonal in the $(\alpha-\beta)$ -plane, contrary to what is said later in the text.

(5) When aiming for interpretability of the results, one would prefer to keep the number of clusters as low as possible to have a really meaningful and interpretable partitioning, with a true variability between each cluster. However, even in the crystalline benchmark where the number of "visually

distinguishable" classes is low (i.e. perfect crystal, vacancy, etc.), having 13 clusters seem unreasonably high for such a trivial example. This is even more striking in the second analysis, where "constructing a small number of useful classes was impossible" (as quoted from the text). Perhaps agglomerative clustering is not the best suited clustering algorithm for that purpose. Have the authors considered using a different approach that penalizes a too high number of clusters (e.g. 10.1198/jcgs.2010.08111) to create a more efficient partitioning?

Finally, a few other comments that I would like the authors to optionally address, but that have a lesser impact on the quality of the manuscript:

(1) The computation of the GIIP distance is apparently implemented in Python and built upon popular frameworks such as PyTorch, which should make anyone able to run it fairly easily. In an effort to promote reproducible research, would the authors agree to make a version of this code publicly available on a platform such as Zenodo or GitHub?

(2) Would the authors agree to mention (without having to show them) a few results on other glassy systems to show the versatility of the method? A comprehensive study of one system is a useful proof of concept, but some glassy systems do not seem to display many interesting structural features (e.g. Nickel-Yttrium alloys), so this would be an interesting test case.

(3) One big question that recent similar works have been trying to address is the predictability of the dynamics of individual atoms using machine-learned clusters or the latent variables derived from high-dimensional descriptors that are yielded by a dimensionality reduction method. Could the authors comment on the applicability and expected performance of their method for such a task?

(4) For the NiNb system, the analysis is restricted to the Ni atoms. Local order usually develops around the smaller component in binary alloys, so one can naturally expect a larger difference between different LAEs for this atomic species. Could the authors comment on the nature and interpretability of the clusters and diffusion coordinates found for the larger atoms?

(5) In addition to the scatter plots (Fig. 2c and 2e), it would be interesting to show the probability density distributions (and the corresponding marginal distributions, perhaps even conditioned to the cluster indices) in the diffusion coordinate planes, to see if they have a certain degree of multimodality (similar to Fig. 3c). This would show if a system clearly favors certain arrangements over others (multimodal) or if the transition between different structures is mostly continuous (unimodal).

(6) For the NiNb glass: is there a clear difference in the resulting diffusion coordinates when training separately on samples from each quench rate (i.e. 4 different sets of diffusion coordinates instead of one)?

Reviewer 1

Reviewer 1, Comment Block 1: Improvement needed in definition of terms

The article may contain useful information. However, the manuscript is very difficult to read due to the vague definition of concepts. The meaning of Fig. 1 and 2 cannot be understood from the previous text. Many variables used in text and graphics are not defined above. It is necessary to give a clear definition of all parameters used in the text before actual using them. What is r , x , α , α' , β , A , B , C , D , X ? And what is their physical meaning? Some things become clear from the experimental procedure, but the meaning of many of the quantities used is not understood yet. What specific functions are used as "a and b"?

Response

We appreciate the reviewer's candor—we want our manuscript to be accessible to as many readers as possible. In response to this comment, we very carefully looked through our manuscript, identified each variable, acronym, and abbreviation, and confirmed that they were defined in their first occurrence in both the abstract and the main text. We also constructed (for ourselves) an index of all the symbols used in the paper and discovered several cases where we were using unnecessarily similar symbols to represent different concepts, leading us to modify the symbol, font, or super/subscripting for many variables. These changes, which improve the clarity of the paper without altering its logic, are too numerous to list exhaustively in this response, so we direct reviewers to the marked-up manuscript.

Reviewer 1, Comment Block 2: Improvement needed in introduction of LAE abstraction

The authors state: "Let X be an atomic configuration, consisting of a set of vectors pointing to atomic positions." How many vectors are used? Where are they pointing? To the nearest atomic neighbors? How is it determined how many vectors are needed for each cluster? How are they recorded? In the form of a matrix?

Response

We have modified and extended the text in question to address this question:

Let K be an LAE, consisting of a set of vectors pointing to atomic positions, and a corresponding set of species for each atom; while these could be stored as a matrix of positions and a vector of species, we note that the atoms in an LAE are unordered. We place no restrictions on the size or nature of the LAE, which could be constructed by selecting a particular number of nearest neighbors, or by setting a cutoff radius. The atomic positions might optionally be shifted to center on a focal atomic position.

Reviewer 1, Comment Block 3: Specific concern with Figure 4

What is plotted along the abscissa x-axis in Fig. 4?

Response

Position in one-dimensional space. We have modified Figure 4, adding x-axis labels (with our spatial signifier, 'x') and harmonizing notation with that appearing in the manuscript.

Reviewer 1, Comment Block 4: Physical meaning & novelty of work unclear

Any work in the field of physics other than pure mathematics must have a clear physical meaning. In view of the above, I propose to make the necessary changes to the manuscript. Without a clear description of the variables, vectors and functions, further consideration of the work is impossible. Moreover, as a summary the authors state: "Applying this approach to a two-dimensional model crystal and a three-dimensional binary model metallic glass results in parameters interpretable as (i.e. one-to-one with) coordination number, composition, volumetric strain, and local symmetry. In particular, we show that a more slowly quenched glass has a higher degree of local tetrahedral symmetry at the expense of cyclic symmetry". Coordination number, composition, volumetric strain, and local symmetry can be easily obtained from the results of MD modeling by the existing methods. The fact that slowly quenched glass has a higher degree of local tetrahedral symmetry at the expense of cyclic symmetry may be also easily confirmed by the existing methods. What is the prime novelty of the proposed approach then? Please, state it clearly.

Response

We have added an additional paragraph to the introduction of the paper that contextualizes the novelty and value of this manuscript:

Our approach complements recent efforts to use machine learning to understand the structure of metallic glasses and supercooled liquids in general. In {cubuk2016structural} the dimensionality method t-SNE {van2014accelerating} was applied to a radial-only basis function expansion of LAEs, neglecting angular information but nonetheless extracting meaningful trends. In {boattini2020autonomously, boattini2019unsupervised} autoencoders were used to elegantly reduce the dimensionality of rotationally invariant descriptors of angular information (Bond Order Parameters), neglecting radial information. In {paret2020assessing} bond angle distribution and radial distribution functions were fed into a clustering algorithm that identified structural communities within an ensemble of LAEs. Finally, {coslovich2022dimensionality} used both angular-only Bond Order Parameters and radial+angular Smooth Overlap of Atomic Positions (SOAP) descriptors in Principle Component Analysis and to train an autoencoder. Taken together, these studies all demonstrate the potential for unsupervised methods to extract meaningful structural information in a disordered material, and highlight the challenge of interpreting dimensionality-reduced structural representations. These studies also share the trait that information was discarded in an initial encoding step preceding dimensionality reduction: radial distributions drop angular information, while Bond Order Parameters drop radial information, are of limited use for analyzing second and higher-order shells of neighbors, and can be discontinuous with respect to atomic

perturbation across a cutoff radius. Even more sophisticated descriptors like SOAP have been shown to systematically drop information {pozdnyakov2020incompleteness}. Information lost in initial encoding necessarily cannot inform a low-dimensional representation. To address this shortcoming, in this manuscript we use a methodology that does not rely on an initial lossy encoding step: we measure generalized distances directly between LAEs, and then apply a dimensionality reduction method that operates on distances rather than on vector encodings. Complementing this advance, we introduce an inter-LAE generalized distance function (the Gaussian Integral Inner Product (GIIP) distance, described in the methods section) that is complete (meaning that the generalized distance between LAEs vanishes if and only if the LAEs are identical; a dimensionality reduction based on an incomplete distance function is necessarily lossy), rotationally invariant, and continuous and smooth with respect to atomic perturbation across a cutoff radius. This combination of qualities is, to our knowledge, novel in the literature.

Reviewer 2

Reviewer 2, Preface

Understanding the microstructure and its evolution in disordered materials is one of long-standing yet the most interesting problems in the field of condensed matter physics. Despite the various models proposed previously in the glass literatures, complete and precise characterization of its atomic-scale microstructure still faces a compelling challenge, particularly for metallic glasses (MGs). Unlike in crystalline materials, it is impossible to extract a unit cell to well describe the microstructure and particularly the structural defects of MGs owing to their lacking of long-range order.

The present manuscript reports on a strategy based on data mining to characterize the microstructural features of amorphous solid materials such as a binary Ni-Nb MG. The outcome of this work paves a useful and effective way to resolve the problem of trade-off between general applicability and interpretability in quantitatively characterizing its atomic-scale structure. Specifically, it seems to be successful in unraveling the atomic short-range ordering and its variation with cooling rate. This part is possibly compelling for some of researchers in the field of MG materials.

Response

It is apparent that Reviewer 2 understands our manuscript, and we thank them for their kind words.

Reviewer 2, Comment Block 1: Limitations of an approach limited to short range correlations

However, this cannot lead to well establish the relationship between properties and microstructure for glassy solid materials. For instance, while the common wisdom suggests that MGs are homogeneous, a large number of experimental and theoretical investigations indicate that they are heterogeneous particularly in a dynamic sense. Therefore, the method proposed in the work in deepening the understanding of the microstructure and properties of MGs is limited. In this case, the reviewer does not recommend it to be published in Nature Communications.

Response

We agree with the limitations highlighted by Reviewer 2: we're not under the illusion that we've cracked the whole structure-properties problem, and we hope that our writing didn't give that impression. We *do* hope that our improved descriptors of local atomic structure will enrich the study of heterogeneity in MG by making spatial variations in glassy structure easier to identify and quantify. We modified our discussion to explicitly acknowledge the importance of heterogeneity and medium-range structural fluctuations:

We note that local structure alone is inadequate to predict the behavior of metallic glasses {egami2023medium, egami2023world}. Our approach does not directly deal with longer-range fluctuations in glassy structure {egami2022medium} that play an important role in e.g. plasticity. However, we hope that improved descriptors of local structure will enhance the conversation around structural heterogeneity by enabling more precise descriptions of the nature of that heterogeneity. We do hypothesize that to some extent the properties of a material will be determined by the volume-averaged distributions of first-nearest-neighbor LAEs in diffusion space---something like a texture

map for a polycrystalline metal---but note that spatial correlations between even fully-characterized LAEs remain difficult to engage. The examples presented in this paper are restricted to first-nearest-neighbor LAEs, but this is not a fundamental limitation of our strategy and it is possible that examination of second- and higher-order neighborhoods will prove illuminating.

Reviewer 2, Question 1

For the two-dimensional crystal, the LAEs could be partitioned into 13 classes with the use of agglomerative clustering. However, for the metallic glass system, the authors claimed “An initial attempt at agglomerative clustering found that the variance in LAEs in the sample was so large...their physical significance”. How did the authors partition the data of the three-dimensional glassy alloy? Are there any rules behind the classification?

Response

We note the similarity of this comment to Reviewer 3’s Specific Comment (5). We believe that Reviewer 2 is asking about the number of clusters in particular, rather than about the rules of agglomerative clustering in general (though, if we misunderstood, we refer the reviewer to “Cluster Analysis” in our methods section).

We have incorporated into the supplementary material new plots showing a histogram of GIIP distances between LAEs and a plot of (number of classes) vs (diameter of largest-diameter classes) for both the XL and MG datasets. These plots show how the XL dataset could be partitioned into 10-13 classes with very low within-class variation (GIIP distance of less than about 0.6), but how agglomerative partitioning of the MG dataset into fewer than 1000 classes resulted in large within-class variation (GIIP distance greater than about 2.0). The usefulness of classes for computational and conceptual purposes drops off significantly when those classes contain substantially dissimilar members, hence our comment quoted by the Reviewer.

In addition to adding supplementary material, we added the following text emphasizing the arbitrary nature of the number of classes:

We note at this point that in an agglomerative clustering scheme the number of classes are ultimately a design decision driven by the purpose of the classification project. For example, if the purpose of LAE classification is as a conceptual research aid, minimizing the number of classes would be of paramount importance (which would in turn drive tolerance of within-class variation). On the other hand, if the purpose of LAE classification is as the basis of an off-lattice kinetic Monte Carlo algorithm, classes would need to contain only LAEs with similar available kinetic pathways, necessitating far less tolerance of within-class variation, and consequently more classes. The trade-off between within-class variation and number of classes is visualized for this two-dimensional crystal example in the supplementary material.

We also modified and added to our formerly very brief description of clustering for the 3d MG sample:

An initial attempt at agglomerative clustering found that for this MG sample, the within-class variation remained high for any number of classes (illustrated in the supplementary

material). In other words, we found little evidence to support the notion that the local structure of metallic glass is amenable to discrete dimensionality reduction. For the purpose of coloring our diffusion space scatter plots, we partitioned the data into twenty classes using agglomerative clustering but make no claim as to their physical significance.

Reviewer 2, Question 2

The authors found that the LAEs plotted in (BCC)-plane of diffusion space (Fig. 2d) locate in a roughly triangular space, with three corners representing tetrahedral, cyclic, and prismatic symmetries, respectively. What are other planes of diffusion space look like? are there similar corners representing the symmetry information of the sample in these planes? If not, why only the (BCC)-plane of diffusion space?

Response

We refer the reviewer to Figure 2(e), which shows the AxD plane of the space, which also has sharp corners. We don't know yet whether those corners represent symmetry conditions, but we think that's a good hypothesis.

We respectfully hope that Reviewer 2 will let us defer this question to our next paper exploring these ideas in greater depth than a Nature Comms paper will permit.

Reviewer 2, Question 3

Are the parameters found in this work, e.g., the one that could be interpreted as local symmetry, applicable to other NiNb alloys or other alloy systems?

Response

Again, we hope that Reviewer 2 will let us defer this question to our next paper exploring these ideas; and while we hold the response to this question out-of-scope for this manuscript, we will share with the reviewer that our current work strongly suggests that the answer is yes for both other NiNb compositions and for other alloy families.

Reviewer 2 is asking the right questions and we are delighted that our manuscript elicited them!

Reviewer 3

Reviewer 3, Preface

I apologize in advance to the authors for the length of this review. I tried to rigorously highlight the strengths and weaknesses of this work, while providing references, directions to consider, as well as optional requirements that I think would make this work more solid.

In this work, the authors presented a new metric, the Gaussian Integral Inner Product (GIIP) distance, that is used to directly assess the similarity or dissimilarity of a pair of local atomic environments (LAEs). An LAE is defined in terms of a smooth atomic density function (i.e. a weighted superposition of Gaussian functions) inspired by the SOAP formalism, which makes it robust to thermal fluctuations and to the definition of nearest neighbors. The GIIP distances between a large number of LAEs can then be used to partition them into a certain number of classes (clusters) by using an agglomerative clustering algorithm. These clusters are composed of similar LAEs, ideally leading to an interpretable partitioning of the system based on its structure. In addition to this partitioning, the GIIP distance can also be used to learn a manifold embedded in a high-dimensional space onto which the LAEs are distributed, using diffusion maps, a non-linear dimensionality reduction method. Being defined by a reduced number of latent variables (called "diffusion coordinates"), this manifold provides a lower-dimensional description of the structure of the system. Some of these diffusion coordinates are highly correlated to standard and well-known observables and order parameters, which provides a useful description of the heterogeneity of the system by using a limited number of variables.

This approach is first illustrated using an interpretable two-dimensional crystalline benchmark, for which the identified clusters are easily identifiable (i.e. crystal, vacancy, etc.) and some of the diffusion coordinates strongly correlated to known observables (coordination and volumetric strain).

After this useful proof of concept, the authors apply their method to a more challenging system: a three-dimensional binary metallic glass. Even though the results are now more difficult to interpret, as one can expect, they still present some interesting features: the main diffusion coordinate strongly correlates with chemical composition, and the distributions of the clusters in the reduced spaces formed by pairs of diffusion coordinates are strongly correlated to different symmetries of the LAE (cyclic, prismatic, etc.).

Response

We appreciate Reviewer 3's thorough and thought-provoking comments, which (we are confident) took considerable time and effort to compose. Reviews like this one do credit to the peer-review system.

Reviewer 3, Comment Block 1: Clarifying novelty and contextualization of this work

Before moving on to a more technical discussion about the methods and the results, I would first like to briefly summarize my general impression of this work: I find the manuscript well-written, clear, and organized in a pleasant way. In particular, I really appreciate the crystalline benchmark, which really helps understanding the approach with clear visual explanations, as it makes the understanding of the second analysis easier. The GIIP distance is a well-defined, elegant and easily interpretable metric to compare LAEs: expressing the difference between two LAEs in units of "atoms of difference" is very

intuitive. Finally, as will be discussed later in the review, several previous works have used dimensionality reduction methods and/or clustering algorithms in similar contexts. However, to the best of my knowledge, diffusion maps is a method that has not yet been used to study the local structure of glasses. I think that this is an interesting and powerful method that should be explored further in future work. And to conclude on the method itself: even though the results of this approach require post-hoc interpretation, this should obviously be contrasted to the commonly called "black-box" nature of many machine-machine learning methods employed in some other works. The results of this study stay reasonably interpretable, with minimal effort, which is what machine learning methods should aim for in my opinion.

Concerning the novelty of this work: while the method itself is new and interesting, I do not find the approach particularly original unfortunately. Reasonably interpretable descriptors (to be contrasted with e.g. SOAP), dimensionality reduction methods and/or clustering algorithms have been used in the past in similar contexts (see e.g. 10.1021/acs.jpcc.6b02144, 10.1038/s41467-020-19286-8, 10.1063/1.5118867, 10.1063/5.0004732, 10.1063/5.0128265) [N.B. in an effort to be fully transparent: this includes some of my recent work, but this does not impact my review in any way, as discussed with the editor]. The general conclusions of the most recent of these works are also similar: latent variables generally correlate well with well-known observables, and clusters can sometimes provide a good insight on the variability of local structures, but they can also be difficult to interpret (especially if there is a large number of them). While there are obvious differences, there are also obvious similarities, and I believe that this study is unfortunately not contextualized enough, as the authors might not know about these previous works. They should try to recontextualize the methods and results, stress the differences, and emphasize on the strengths and weaknesses before it is suitable for publication.

Response

We appreciate Reviewer 3's willingness to be transparent about potential conflicts of interest. We have incorporated the citations suggested by Reviewer 3 in a new introductory paragraph that better contextualizes the novelty and value of our manuscript:

Our approach complements recent efforts to use machine learning to understand the structure of metallic glasses and supercooled liquids in general. In {cubuk2016structural} the dimensionality method t-SNE {van2014accelerating} was applied to a radial-only basis function expansion of LAEs, neglecting angular information but nonetheless extracting meaningful trends. In {boattini2020autonomously, boattini2019unsupervised} autoencoders were used to elegantly reduce the dimensionality of rotationally invariant descriptors of angular information (Bond Order Parameters), neglecting radial information. In {paret2020assessing} bond angle distribution and radial distribution functions were fed into a clustering algorithm that identified structural communities within an ensemble of LAEs. Finally, {coslovich2022dimensionality} used both angular-only Bond Order Parameters and radial+angular Smooth Overlap of Atomic Positions (SOAP) descriptors in Principle Component Analysis and to train an autoencoder. Taken together, these studies all demonstrate the potential for unsupervised methods to extract meaningful structural information in a disordered material, and highlight the challenge of interpreting dimensionality-reduced structural representations. These studies also share the trait that information was discarded in an initial encoding step

preceding dimensionality reduction: radial distributions drop angular information, while Bond Order Parameters drop radial information, are of limited use for analyzing second and higher-order shells of neighbors, and can be discontinuous with respect to atomic perturbation across a cutoff radius. Even more sophisticated descriptors like SOAP have been shown to systematically drop information {pozdneyakov2020incompleteness}. Information lost in initial encoding necessarily cannot inform a low-dimensional representation. To address this shortcoming, in this manuscript we use a methodology that does not rely on an initial lossy encoding step: we measure generalized distances directly between LAEs, and then apply a dimensionality reduction method that operates on distances rather than on vector encodings. Complementing this advance, we introduce an inter-LAE generalized distance function (the Gaussian Integral Inner Product (GIIP) distance, described in the methods section) that is complete (meaning that the generalized distance between LAEs vanishes if and only if the LAEs are identical; a dimensionality reduction based on an incomplete distance function is necessarily lossy), rotationally invariant, and continuous and smooth with respect to atomic perturbation across a cutoff radius. This combination of qualities is, to our knowledge, novel in the literature.

Reviewer 3, Specific Comment (1.a): Why DMaps vs e.g. PCA?

How does diffusion maps compare to basic PCA? Manifold learning can be very useful when the high-dimensional distribution to study is complex and intricate, but in the case of disordered systems like glasses, one can expect a reasonably homogeneous and unimodal distribution (see 10.1063/5.0128265). In such a case, complex dimensionality reduction methods like autoencoders do not seem to perform particularly better than standard PCA, since there is no complex manifold to learn (i.e. only a simple projection from a high-dimensional space onto a lower-dimensional hyperplane).

Response

This is a great question, and while some evidence (e.g. the reference provided) suggests that certain disordered systems may have a reasonably homogeneous and unimodal distributions of structural features, it is certainly not definitively established that this is so. Even some of the systems studied in the reference exhibit a degree of bimodality, which suggests a set of competing underlying structural forms. Moreover, our analysis started with an objective that differed in subtle ways from previous works. In established formalisms for learning structural descriptors, i.e. SOAP and SBO, the initial objective is to directly obtain a vector of descriptors from a reduced representation of the atomic neighborhood (i.e. using spherical harmonics). Our initial objective, on the other hand, was to establish a means of computing **distance** between two **complete** local atomic neighborhoods in a manner that is rotationally and permutation invariant. In other words, if we're looking at two atomic neighborhoods, can we establish a measure of how different those neighborhoods are? Digging deeper into this question, it becomes immediately clear that such a meaningful distance cannot be established in Euclidean space. The reasons for this arise from the canonical ordering problem, which essentially states that any Euclidean distance measure between local atomic neighborhoods (i.e. the positions of the atoms) will be highly dependent on the ordering of the atoms in the list and there is no identifiable

ordering that can produce a consistent distance measure between neighborhoods. In other words, the distance measure cannot be made permutationally invariant.

Our second objective was to establish a distance measure that was rooted in a complete description of the local atomic neighborhoods (not a reduced representation, i.e. spherical harmonics). The complete description can be established by the atomic positions (and perhaps some other information, e.g. atomic radius, potential energy, etc. – more on this later) in Euclidean space. But, per the discussion above, we cannot build a meaningful distance in Euclidean space. So we smear the local atomic neighborhood using a series of Gaussians, which allows us to encode the complete atomic neighborhood (and also encode other features such as radius, PE, etc. by modifying the parameters of Gaussians). With this continuous functional description of the environment, we can now establish a distance in ***the space of these functions***. This is the basis for our derived Gaussian Integral Inner Product (GIIP) distance.

Now that the distance is established in function space, it becomes immediately clear that PCA is not an option. PCA inherently operates in Euclidean spaces, but our representation no longer resides in a Euclidean space. Note: we could discretize the function and bring it back to Euclidean space but we again lose the meaningfulness of our distance measure. Given the established distance measure, the natural fit (at least as we saw it) was to apply kernel-based dimension reduction in which the distance measure could be directly applied within a kernel; hence our choice of applying diffusion maps. Note, we could have used other kernel-based methods such as kernel PCA, but we wouldn't expect the results to differ drastically.

To conclude, we believe that looking at the problem in this slightly different way provides some additional insight into local atomic structure that is perhaps not captured using linear methods based on reduced representations. In particular, it allows us to establish a physically meaningful (and intuitive) direct measure of distance (similarity/dissimilarity) between complete local atomic neighborhoods that can then be used to establish reduced dimensional descriptors that we describe local atomic structure in interesting and potentially useful ways.

Reviewer 3, Specific Comment (1.b): Comparison of DMaps vs others in general

How does diffusion maps compare to other non-linear dimensionality reduction methods, that can also learn a complex manifold to project new samples onto it (e.g. neural network autoencoders, locally linear embedding, etc.)?

Response

Building from the previous question, we recognize that other non-linear dimension reduction methods could be applied. As already mentioned, we could use other kernel-based methods such as kernel PCA or we could use methods such as variational autoencoders (VAEs). The application of these other approaches is beyond the scope of our investigation but is definitely of interest to us. While we don't expect other kernel-based methods would result in drastically different encodings, VAEs (and other encoder architectures) would be very interesting to study. That said, they operate differently than the proposed approach and, again, do not give us the tangible distance measure we seek so they are not considered in this study.

Reviewer 3, Specific Comment (2): Concerns about bias in training set construction

I would appreciate a more detailed explanation on how the training sets are built. For instance, for the crystalline benchmark, only atoms at the two extremes of the potential energy distribution $p(U)$ are selected. Consequently, the training set is not statistically representative of the whole dataset since its elements are "hand-picked", which is generally not an intuitive choice when building a training set. Could the authors explain the motivation behind this choice?

Response

We understand that a hand-picked training set such as the one cited in the comment would invalidate or degrade many analyses, including PCA. However, a "hand-picked" dataset can be acceptable for the specific task of parameterizing a nonlinear manifold.

Imagine a two-dimensional manifold—a curved surface—in three-space, sampled by datapoints. Those datapoints might be sampled more or less densely in regions of the manifold. There might be one spot on the manifold where points are very densely clustered indeed. Now, imagine randomly dropping 90% of the datapoints from the dataset. The resulting sparsified dataset is statistically representative of the original dataset, but there might be gaping holes left in the sparsely-sampled regions of the manifold; and Diffusion Maps cannot learn to parameterize portions of the manifold that are not adequately sampled. On the other hand, imagine taking the original sampling of the manifold and dropping 90% of the data, but selectively dropping that data from the very densely sampled regions of the manifold. Looking at the sparsified manifold now, it retains its original shape and the Diffusion Maps algorithm can parameterize the full manifold without trouble. The particular parameterization obtained might vary from the one that Diffusion Maps would generate from the full dataset, but in this second case the correct manifold is being parameterized.

Our sampling strategy for the two-dimensional crystal follows from this insight: that environments near the ground state are very common and are all very similar, and are therefore not very useful in conveying the shape of the manifold. On the other hand, higher-energy environments are rare and vary greatly, and that is precisely the variation that we wish to capture. We deliberately retain these high-energy environments for our training set because they are the most valuable for learning the shape of the manifold.

At no point do we make any claims that would rely on the distribution of the training and extension sets being statistically similar: we only claim that we learned the manifold on which the LAEs of the material fall.

Reviewer 3, Specific Comment (3): Cutoffs and significance vs redundancy

I found the "significance vs. redundancy" approach to select the diffusion coordinates very interesting. It is more interpretable and easier to justify than, e.g. a rule of thumb on the amount of explained variance with PCA. However, how is the final number of diffusion coordinates chosen? Is there a well-defined criterion or just a visual estimation based on this scatter plot?

Response

We added the following text to the section on the two-dimensional crystal example:

In this simplified case it was clear that the first four diffusion coordinates are more valuable than the rest. However, for other materials, selecting the low-dimensional set of diffusion coordinates might be more arbitrary as redundancy and eigenvalue fade smoothly to 1 and 0, respectively. In these cases, thresholds for redundancy and importance would have to be selected that balance simplicity against detailed descriptiveness in view of the task at hand, in the same spirit as setting the number of classes.

Reviewer 3, Specific Comment (4): Realignment and orthogonality in Fig. 1

In Fig. 1(c), having to "redefine"/"realign" the latent variables into α' and β' so they better correlate with interpretable quantities (coordination and strain) seems a bit artificial. One should not have to "tweak" the latent variables: they are learned in an unsupervised way in the first place for that reason, so they can be directly used to learn about the structure. Also, α' and β' do not appear to be orthogonal in the $(\alpha-\beta)$ -plane, contrary to what is said later in the text.

Response

Would that the math worked that way. The Diffusion Maps algorithm produces a parameterization of a structural manifold that is rotated or distorted depending on the specifics of how the data was sampled; because DMaps operates on distances alone, it knows nothing of the physics that produced those distances, so the resulting parameterization might or might not be neatly aligned with physical observables.

Fortunately, parameterizations can be rotated or bijectively distorted and remain valid parameterizations of the same manifold, which is what we did here. That α' and β' do not appear orthogonal in Figure 1(c) is a consequence of the axis scaling. We have adjusted the axis scaling so that they appear orthogonal to avoid confusion on this point. We have also added a citation {evangelou2022parameter} to help guide interested readers through the thought process.

Unraveling this one step further for the sake of completeness—in a setting where axes can be scaled and distorted freely, the entire notion of orthogonality becomes a bit arbitrary. A more complete description in our paper might have been "orthogonal in the normalized eigenframe returned by the diffusion maps algorithm" but that felt like too much for most readers.

Reviewer 3, Specific Comment (5):

When aiming for interpretability of the results, one would prefer to keep the number of clusters as low as possible to have a really meaningful and interpretable partitioning, with a true variability between

each cluster. However, even in the crystalline benchmark where the number of "visually distinguishable" classes is low (i.e. perfect crystal, vacancy, etc.), having 13 clusters seem unreasonably high for such a trivial example. This is even more striking in the second analysis, where "constructing a small number of useful classes was impossible" (as quoted from the text). Perhaps agglomerative clustering is not the best suited clustering algorithm for that purpose. Have the authors considered using a different approach that penalizes a too high number of clusters (e.g. 10.1198/jcgs.2010.08111) to create a more efficient partitioning?

Response

We note the similarity of this comment to Reviewer 2's Question 1. We have incorporated into the supplementary material new plots showing a histogram of GIIP distances between LAEs and a plot of (number of classes) vs (diameter of largest-diameter classes) for both the XL and MG datasets. These plots show how the XL dataset could be partitioned into 10-13 classes with very low within-class variation (GIIP distance of less than about 0.6), but how agglomerative partitioning of the MG dataset into fewer than 1000 classes resulted in large within-class variation (GIIP distance greater than about 2.0). The usefulness of classes for computational and conceptual purposes drops off significantly when those classes contain substantially dissimilar members.

In addition to adding supplementary material, we added the following text emphasizing the arbitrary nature of the number of classes:

We note at this point that in an agglomerative clustering scheme the number of classes are ultimately a design decision driven by the purpose of the classification project. For example, if the purpose of LAE classification is as a conceptual research aid, minimizing the number of classes would be of paramount importance (which would in turn drive tolerance of within-class variation). On the other hand, if the purpose of LAE classification is as the basis of an off-lattice kinetic Monte Carlo algorithm, classes would need to contain only LAEs with similar available kinetic pathways, necessitating far less tolerance of within-class variation, and consequently more classes. The trade-off between within-class variation and number of classes is visualized for this two-dimensional crystal example in the supplementary material.

We also modified and added to our formerly very brief description of clustering for the 3d MG sample:

An initial attempt at agglomerative clustering found that for this MG sample, the within-class variation remained high for any number of classes (illustrated in the supplementary material). In other words, we found little evidence to support the notion that the local structure of metallic glass is amenable to discrete dimensionality reduction. For the purpose of coloring our diffusion space scatter plots, we partitioned the data into twenty classes using agglomerative clustering but make no claim as to their physical significance.

We also, on reflection, reduced the number of classes for the XL case to 10 to maximize human-interpretability, but this is ultimately as arbitrary as 13 was.

Reviewer 3, Optional Comment (1): Open-Source Codebase

The computation of the GIIP distance is apparently implemented in Python and built upon popular frameworks such as PyTorch, which should make anyone able to run it fairly easily. In an effort to promote reproducible research, would the authors agree to make a version of this code publicly available on a platform such as Zenodo or GitHub?

Response

Yes. We are in the process of getting national laboratory approval for open-sourcing the GIIP code. We will incorporate a link in the final manuscript. We have also included a copy of our data and source code with this revision submission and assume that the editor can pass it along to you if you are interested.

Reviewer 3, Optional Comment (2): Mentioning other glassy systems

Would the authors agree to mention (without having to show them) a few results on other glassy systems to show the versatility of the method? A comprehensive study of one system is a useful proof of concept, but some glassy systems do not seem to display many interesting structural features (e.g. Nickel-Yttrium alloys), so this would be an interesting test case.

Response

We are currently working on a follow-on manuscript that analyzes the full space of MGs for which we have trustworthy (within reason) interatomic potentials. We hope that Reviewer 3 will allow us to defer the answer to this question to that paper.

Reviewer 3, Optional Comment (3): GIIP descriptors as predictors of short-range dynamics

One big question that recent similar works have been trying to address is the predictability of the dynamics of individual atoms using machine-learned clusters or the latent variables derived from high-dimensional descriptors that are yielded by a dimensionality reduction method. Could the authors comment on the applicability and expected performance of their method for such a task?

Response

We have added to the discussion:

One potential use of learned descriptors is as predictors of the local kinetic pathways available to atoms {cubuk2016structural}. To the extent that fine-grained atomic behavior is predictable in MGs (see {zella2022transient}), our GIIP-based descriptors are a potential (if very slow) approach for reducing the dimensionality of the predictor domain.

Reviewer 3, Optional Comment (4): The Nb half of the system

For the NiNb system, the analysis is restricted to the Ni atoms. Local order usually develops around the smaller component in binary alloys, so one can naturally expect a larger difference between different LAEs for this atomic species. Could the authors comment on the nature and interpretability of the clusters and diffusion coordinates found for the larger atoms?

Response

This is a computationally expensive method due to the brute-force optimization over $O(3)$ in Eqn. 10; consequently we economized by running only a small-scale ($n=3000$ LAEs) analysis on Nb-centered LAEs and focusing our resources the Ni-centered LAEs where we expected more interesting structure to emerge. We withheld the results of our small-scale analysis from the article because of doubts over whether 3000 LAEs were representative of the glass, but the study did strongly suggest that LAEs centered on Nb atoms are collectively less structured than Ni-centered LAEs. We could revisit the decision to withhold the Nb-centered results if Reviewer 3 insists.

Reviewer 3, Optional Comment (5): Probability density distributions, favoring arrangements

In addition to the scatter plots (Fig. 2c and 2e), it would be interesting to show the probability density distributions (and the corresponding marginal distributions, perhaps even conditioned to the cluster indices) in the diffusion coordinate planes, to see if they have a certain degree of multimodality (similar to Fig. 3c). This would show if a system clearly favors certain arrangements over others (multimodal) or if the transition between different structures is mostly continuous (unimodal).

Response

A good idea. In fact, we have plotted Fig. 2(c) with points colored by kernel density estimate to illustrate that, yes, it is multimodal (as one would expect given that atoms are discrete). We have updated the caption to reflect this.

We have added versions of Fig. 2(d) and Fig. 2(e) colored by kernel density estimate to the supplementary material.

Reviewer 3, Optional Comment (6):

For the NiNb glass: is there a clear difference in the resulting diffusion coordinates when training separately on samples from each quench rate (i.e. 4 different sets of diffusion coordinates instead of one)?

Response

Following from our discussion of manifold parameterization above, we wouldn't expect the 4 sets of diffusion coordinates to be identical, but we would hope for them to parameterize the same manifold. We calculated the 4 sets of diffusion coordinates, each trained on only a single quench, but extended (see our brief discussion of the Nystrom extension in the paper) across all four quenches. We then used multivariable polynomial regression to analyze whether these sets of diffusion coordinates were mutually bijective. To our relief, we are indeed able to map from one set of diffusion coordinates to the low-order terms of the other sets of diffusion coordinates with all adjusted R-squared values greater than 0.997.

This strongly suggests that training separately on single quench rates produces diffusion maps parameterizing the same manifold, i.e. encoding the same information. Thanks for suggesting this mini-experiment!

REVIEWER COMMENTS

Reviewer #1 (Remarks to the Author):

In the corrected form, the article looks much better, and in general, deserves publication in Nature Communication. The methodology has become clearer. Also, the value of this technique in application to the search for dislocations in two-dimensional crystals is well understood. However, what about real 3D crystals? Why is this technique not tested on defects in them? Is there any difficulty here?

Nevertheless, I believe that the article does not fully explain what is the practical value of this work as applied to three-dimensional metallic glasses (except for the original methodology)? The presence of clusters of different symmetry and chemical composition was shown. But, is it possible to quantify their fraction in relation to the total number of clusters? For example, the Voronoi polyhedra analysis makes it possible to indicate the presence of clusters of a certain symmetry, as well as the relative fraction of some clusters (of some symmetry, composition, coordination number) in relation to the others. Each cluster is also uniquely tied to spatial coordinates. This technique allows a detailed comparison of the structures of NiNb glasses obtained at different cooling rates. By analogy, this work would be very adorned by the possibility of determining the quantitative fraction of clusters of a certain symmetry, if possible.

Reviewer #2 (Remarks to the Author):

After the review of the rebuttals and revisions on the manuscript, I do see the authors have replied well most concerns of reviewers. The manuscript is somehow improved. However, I still have one suggestion: short range order or middle range order in amorphous materials is very sensitive to any heating, which induce some diffusion of atoms in the material. Moreover, according to numerous experimental and computational results, the inherent features of amorphous condense matters including MGs also strongly depend on chemical composition of glassy alloys. If the author can provide some solid information evidences that the LEA approach can describe as well the chemical dependence of short-range order in a MG system such as Ni_{100-x}Nb_x or Cu_{100-x}Zr_x, both of which have been well previously studied as model materials to understand topological and/or chemical atomic-scale evolution of MGs with chemical composition, I may recommend accepting this paper.

Reviewer #3 (Remarks to the Author):

I appreciate the authors' efforts in addressing the questions raised during the initial review. I have carefully examined the revised manuscript and evaluated their responses to the initial review comments: most of the changes suggested by the reviewers have been successfully incorporated and the authors have answered the remaining points in a clear and convincing way.

In their revision, the authors have done a good job at recontextualizing their study by looking deeper at the existing literature. The explanations on the novelty and significance of the method developed in the paper are clear and pedagogical.

I also greatly appreciate the addition of the supplemental information, as it provides more details and clarifications that support the main findings and conclusions presented in the paper.

The authors' decision to make their code publicly available (upon approval of the national laboratory) also contributes to the transparency of their study and is a nice effort to promote reproducible research. I hope this will inspire other researchers to adopt similar practices.

Finally, I would say that no additional analysis is needed for the publication of the paper, as the main concerns from the reviewers have been addressed. However, I would suggest that the revised manuscript could benefit from incorporating some of the comments made in the rebuttal letter to the referees, for instance:

- The explanation on the construction of the training dataset. A less detailed explanation or a reference to a relevant article on parameterizing non-linear manifolds would prevent other people who, like myself, would expect a uniform sampling for such a task.
- A brief comment on the study conducted for the Nb atoms. While I do not believe that showing these results would be necessarily relevant (due to the fact that, as expected, they are "collectively less structured than Ni-centered LAEs"), showing that this part of the study has not been completely discarded would be beneficial.
- Similarly, a brief comment on the diffusion coordinates parameterizing the same manifold when trained on different quench rates. I personally find this result quite interesting and non-trivial. These are obviously not critical, but I believe these minor additions could enhance the accessibility of the paper and demonstrate the meticulous efforts invested in analyzing the finest details.

Overall, I believe that the revised manuscript has significantly improved since its previous version and makes valuable contributions to the field. I recommend accepting this paper for publication.

Reviewer 1

Reviewer 1, Comment 1: Better, but why not 3d crystals?

In the corrected form, the article looks much better, and in general, deserves publication in Nature Communication. The methodology has become clearer. Also, the value of this technique in application to the search for dislocations in two-dimensional crystals is well understood. However, what about real 3D crystals? Why is this technique not tested on defects in them? Is there any difficulty here?

Response

We appreciate Reviewer 1 taking the time to reevaluate our work—their suggestions in the previous round were critical to the revision process. There is no special difficulty in applying our approach to 3d crystals to e.g. identify dislocations. However, there are already better tools than ours for identifying common crystal defects, so we restricted our writeup to an easy-to-visualize pedagogical case (the 2d crystal) before jumping to the far more difficult 3d glass (for which the preexisting toolset leaves much to be desired). We have added the following text to the concluding paragraph of the 2d crystal section:

In principle, the methods described in this section could readily be applied to a three-dimensional crystal to extract defects, but in our view there are already widely-used better-suited tools for this task.

Reviewer 1, Comment 2: Numbers of LAEs in various symmetries

Nevertheless, I believe that the article does not fully explain what is the practical value of this work as applied to three-dimensional metallic glasses (except for the original methodology)? The presence of clusters of different symmetry and chemical composition was shown. But, is it possible to quantify their fraction in relation to the total number of clusters? For example, the Voronoi polyhedra analysis makes it possible to indicate the presence of clusters of a certain symmetry, as well as the relative fraction of some clusters (of some symmetry, composition, coordination number) in relation to the others. Each cluster is also uniquely tied to spatial coordinates. This technique allows a detailed comparison of the structures of NiNb glasses obtained at different cooling rates. By analogy, this work would be very adorned by the possibility of determining the quantitative fraction of clusters of a certain symmetry, if possible.

Response

That's a good comment. In our last revision we added some supplementary material that showed the kernel density estimate maps for the 3d glass in the BxC and AxD planes; these maps theoretically allow quantitative comparisons of the numbers of LAEs populating the various symmetries. However, your comment leads us to believe that it would be helpful for us to integrate the KDE maps and provide actual fractions of LAEs in the various symmetry sectors, so we have added the following two figures to the SI expanding on the quantitative fractions of LAEs in each symmetry sector of the BxC plane, along with a short addition to the main text pointing to them:

Populations of cyclic, tetrahedral, and prismatic sectors of the BxC plane as a function of cooling rate. These sectors are defined in the next figure.

Division of the BxC plane into tetrahedral, cyclic, and prismatic sectors using a 3-cluster Gaussian Mixture model; the aggregated populations of these regions are shown in the figure above for the four different quench rates.

Reviewer 2

Reviewer 2, Preface

After the review of the rebuttals and revisions on the manuscript, I do see the authors have replied well most concerns of reviewers. The manuscript is somehow improved. However, I still have one suggestion: short range order or middle range order in amorphous materials is very sensitive to any heating, which induce some diffusion of atoms in the material. Moreover, according to numerous experimental and computational results, the inherent features of amorphous condense matters including MGs also strongly depend on chemical composition of glassy alloys. If the author can provide some solid information evidences that the LEA approach can describe as well the chemical dependence of short-range order in a MG system such as Ni_{100-x}Nb_x or Cu_{100-x}Zr_x, both of which have been well previously studied as model materials to understand topological and/or chemical atomic-scale evolution of MGs with chemical composition, I may recommend accepting this paper.

Response

Addressing the first concern, we agree that the paper (and, for that matter, our peers studying the structure of metallic glass) would benefit from a little more acknowledgement of the transient nature of SRO and MRO in the glass in the presence of thermal (and, probably to some extent, quantum mechanical) effects. We have added the following:

Stepping back, however, we note the dynamic nature of both short-range and medium-range order in metallic glasses at finite temperature \cite{egami2022medium, egami2023medium}, the chaotic nature of which may defeat the predictive power of static structural descriptors in general.

Addressing the second concern, Reviewer 2 has again raised precisely the sorts of questions that we hope to address using this methodology. We have been working diligently since initially submitting this article to survey the space of binary metallic glasses using atomistic simulation, building towards a longer, more comprehensive follow-on manuscript. We hope that Reviewer 2 will allow us to hold their suggested analysis out-of-scope for this paper, in anticipation of a thorough response in the near future.

We regret that the peer-review process is blind, because we suspect we would enjoy a conversation with Reviewer 2 about these and adjacent issues. Perhaps our paths will cross at a conference this year. Thank you for taking the time to consider our work!

Reviewer 3

Reviewer 3, Preface

I appreciate the authors' efforts in addressing the questions raised during the initial review. I have carefully examined the revised manuscript and evaluated their responses to the initial review comments: most of the changes suggested by the reviewers have been successfully incorporated and the authors have answered the remaining points in a clear and convincing way.

In their revision, the authors have done a good job at recontextualizing their study by looking deeper at the existing literature. The explanations on the novelty and significance of the method developed in the paper are clear and pedagogical.

I also greatly appreciate the addition of the supplemental information, as it provides more details and clarifications that support the main findings and conclusions presented in the paper.

The authors' decision to make their code publicly available (upon approval of the national laboratory) also contributes to the transparency of their study and is a nice effort to promote reproducible research. I hope this will inspire other researchers to adopt similar practices.

Finally, I would say that no additional analysis is needed for the publication of the paper, as the main concerns from the reviewers have been addressed. However, I would suggest that the revised manuscript could benefit from incorporating some of the comments made in the rebuttal letter to the referees, for instance: <Itemized below>

These are obviously not critical, but I believe these minor additions could enhance the accessibility of the paper and demonstrate the meticulous efforts invested in analyzing the finest details.

Overall, I believe that the revised manuscript has significantly improved since its previous version and makes valuable contributions to the field. I recommend accepting this paper for publication.

Response

We thank Reviewer 3 for their kind words and continue to be grateful for their own anonymous contributions in the current and previous rounds of reviews. We have incorporated the suggestions here in full.

Reviewer 3, Suggested Inclusion 1: Non-uniform sampling of dataset

The explanation on the construction of the training dataset. A less detailed explanation or a reference to a relevant article on parameterizing non-linear manifolds would prevent other people who, like myself, would expect a uniform sampling for such a task.

Response

We have added a footnote:

A hand-picked training set such as the one described above would invalidate or degrade many analyses, but can be acceptable for the specific task of parameterizing a nonlinear manifold. The goal of the training set is not to be statistically representative of the

particular sampling of the manifold in the full dataset; rather, the goal is for the training set to thoroughly sample the shape of the manifold. Tight clusters of nearly identical samples---such low-energy crystalline configurations in this example---convey very little information about the manifold, whereas the higher-energy defective states are spread across the manifold in a very useful way.

Reviewer 3, Suggested Inclusion 2: Nb-centered LAEs

A brief comment on the study conducted for the Nb atoms. While I do not believe that showing these results would be necessarily relevant (due to the fact that, as expected, they are "collectively less structured than Ni-centered LAEs"), showing that this part of the study has not been completely discarded would be beneficial.

Response

We have added a footnote:

We also ran a small-scale (n=3000) analysis on Nb-centered LAEs, but focused our computational resources on Ni-centered LAEs when we found that more interesting structure emerged around Ni atoms. We withhold the results of our small-scale Nb-centered analysis due to space constraints and doubts over whether 3000 LAEs are representative of the glass.

Reviewer 3, Suggested Inclusion 3

Similarly, a brief comment on the diffusion coordinates parameterizing the same manifold when trained on different quench rates. I personally find this result quite interesting and non-trivial.

Response

We have added to the end of the section on quench rates:

As a final note, in an effort to determine whether diffusion coordinates trained on individual quench datasets parameterized the same manifold, we calculated four sets of diffusion coordinates, with each set trained on a single quench but extended across all four quenches. We then used multivariable polynomial regression to analyze whether these four sets of diffusion coordinates were mutually bijective. We found that we were able to regress from each set of diffusion coordinates to the low-order terms of the other sets of diffusion coordinates with all adjusted R^2 values greater than 0.997. This strongly suggests that training separately on single quench rate datasets produces diffusion maps parameterizing the same manifold i.e. encoding the same information.

REVIEWERS' COMMENTS

Reviewer #1 (Remarks to the Author):

The authors revised the manuscript in a proper way. It is ready for acceptance. I only forget to mention that the structure of Ni-Nb glassy/amorphous alloys was studied in detail by neutron scattering:
Journal of Non-Crystalline Solids 46 (1981) 125-134;
Journal of Non-Crystalline Solids 104 (1988) 291-299
and mapped by scanning tunneling microscopy at the atomic scale:
Journal of Alloys and Compounds 816 (2020) 152680.
The results of these works can be used for comparison if the authors wish so. After their final decision on this matter (optional), the manuscript can be accepted for publication.

Reviewer #2 (Remarks to the Author):

Indeed, the authors have well addressed most concerns of reviewers in this revised manuscript, which may be in favor of understanding the amorphous structure of glassy materials with the description method as proposed in the present work. Anyway, the reviewer do wish the outcome of this research could pave an effective way of establish the relationship between the properties and inherent structure for glassy alloys. Consequently, it would be much better for the authors of this work to give more words about the issues which are unresolved via the method in the revised version. At the current situation, the reviewer has no more suggestion concerning the publication of this paper in the Journal of nature communications.

Reviewer #3 (Remarks to the Author):

I am glad to see that all the comments from the reviewers have been carefully addressed, demonstrating a commendable level of commitment to enhancing the paper. I was already satisfied with the previous revision, therefore I welcome these latest changes that improved the overall quality of the paper even further. I believe it is suitable for publication without the need for any additional requests and I am looking forward to seeing any related future work.

Reviewer 1

Reviewer 1: Comparison to Ni-Nb experimental characterization

The authors revised the manuscript in a proper way. It is ready for acceptance.

I only forget to mention that the structure of Ni-Nb glassy/amorphous alloys was studied in detail by neutron scattering:

Journal of Non-Crystalline Solids 46 (1981) 125-134;

Journal of Non-Crystalline Solids 104 (1988) 291-299

and mapped by scanning tunneling microscopy at the atomic scale:

Journal of Alloys and Compounds 816 (2020) 152680.

The results of these works can be used for comparison if the authors wish so. After their final decision on this matter (optional), the manuscript can be accepted for publication.

Response

We are grateful for Reviewer 1's recommendation. We feel that extending our methodology to experimental characterization is a crucial next step (which will appear in a follow-up manuscript), but is outside the scope of this manuscript. However, recognizing the value of comparison to preexisting experimental results, we have included citations to the above-suggested papers in our manuscript. We also point to the STM results referenced to here in the Discussion in connection with Reviewer 2's comment.

Reviewer 2:

Indeed, the authors have well addressed most concerns of reviewers in this revised manuscript, which may be in favor of understanding the amorphous structure of glassy materials with the description method as proposed in the present work. Anyway, the reviewer do wish the outcome of this research could pave an effective way of establish the relationship between the properties and inherent structure for glassy alloys. Consequently, it would be much better for the authors of this work to give more words about the issues which are unresolved via the method in the revised version. At the current situation, the reviewer has no more suggestion concerning the publication of this paper in the Journal of nature communications.

Response

We are grateful for Reviewer 2's willingness to go through this revision process. We have added the following to the Discussion to elaborate on issues that remain unresolved:

There remain open questions around the degree to which the model NiNb metallic glass studied here is structurally similar to physical NiNb glass, and to which the structural insights observed here might extend to other compositions and systems. We look forward to connecting GIIP to a growing body of atomistic experimental data for metallic glass (for example, {belosludov2020atomic}); we also look forward to applying this methodology across a wide range of model metallic glass systems.

Reviewer 3:

I am glad to see that all the comments from the reviewers have been carefully addressed, demonstrating a commendable level of commitment to enhancing the paper. I was already satisfied with the previous revision, therefore I welcome these latest changes that improved the overall quality of the paper even further. I believe it is suitable for publication without the need for any additional requests and I am looking forward to seeing any related future work.

Response

We are grateful for Reviewer 3's recommendation.

To the editor:

Thank you for your time and effort bringing this paper into its final form. While anonymity is integral to the peer review system, it is strange to me that reviewers who contributed so much to this manuscript would not have their names anywhere mentioned. I thankfully acknowledge their time, expertise, and even-handed patience through the revision process.